# Capturing Visual Environment Structure Correlates with Control Performance

**Jiahua Dong**
University of Illinois Urbana-Champaign
jiahuad2@illinois.edu

**Yunze Man**
University of Illinois Urbana-Champaign
yunzem2@illinois.edu

**Pavel Tokmakov**[†]
Toyota Research Institute
pavel.tokmakov@tri.global

**Yu-Xiong Wang**[†]
University of Illinois Urbana-Champaign
yxw@illinois.edu

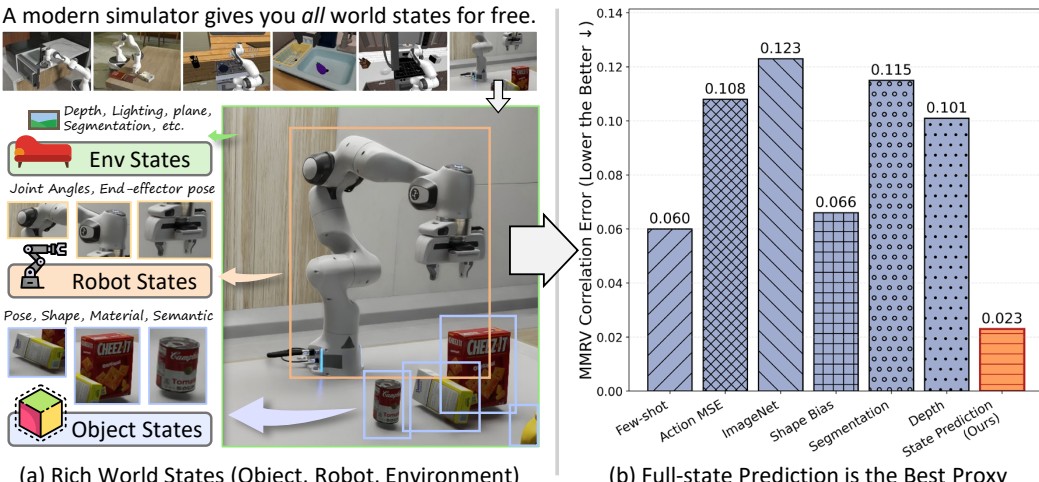

(a) Rich World States (Object, Robot, Environment)  (b) Full-state Prediction is the Best Proxy

Figure 1: Simulation environments provide access to full world state labels for free (left), enabling our proxy task — state prediction from visual inputs. This proxy strongly correlates with downstream policy success across environments and architectures (right, results for SimplerEnv environment).

## ABSTRACT

The choice of visual representation is key to scaling generalist robot policies. However, direct evaluation via policy rollouts is expensive, even in simulation. Existing proxy metrics focus on the representation's capacity to capture narrow aspects of the visual world, like object shape, limiting generalization across environments. In this paper, we take an analytical perspective: we probe pretrained visual encoders by measuring how well they support decoding of environment state — including geometry, object structure, and physical attributes — from images. Leveraging simulation environments with access to ground-truth state, we show that this probing accuracy strongly correlates with downstream policy performance across diverse environments and learning settings, significantly outperforming prior metrics and enabling efficient representation selection. More broadly, our study provides insight into the representational properties that support generalizable manipulation, suggesting that learning to encode the latent physical state of the environment is a promising objective for control. Code and models are available at our project page.

## 1  INTRODUCTION

Although robotics has seen significant progress in developing generalist real-world manipulation policies in recent years (Black et al., 2025; Brohan et al., 2022b; Ghosh et al., 2024), its pace of

---

[†]Equal advising.

advancement remains relatively slow. This is largely due to the prohibitively high cost of policy evaluations in the real world. Most works use simulation environments for model development (Li et al., 2024), but simulation policy rollouts remain significantly more expensive than standard metrics used in vision or natural language processing (NLP). One particularly expensive and influential component of this process is the choice of visual representation, which modern generalist policies heavily rely on (Majumdar et al., 2023a). Recent work (Burns et al., 2024) has explored image-level proxy metrics, such as object segmentation accuracy, as a way to approximate representation quality without requiring full rollouts, but these findings remain limited in scope and generality.

More broadly, the question of what makes a visual representation useful for manipulation has received a lot of attention recently (Burns et al., 2024; Chen et al., 2021; Majumdar et al., 2023a). Among those, a growing body of work has explored reconstruction-based pretraining objectives (Hafner et al., 2020; He et al., 2022a) as visual representations for robotics (Radosavovic et al., 2022; Shridhar et al., 2023; Zhang et al., 2025b). These approaches are rooted in the idea that learning to capture the sparse state of the environment from dense visual observations leads to representations that are more aligned with the needs of control. Very recently, Qi et al. (2025) provide further support of this hypothesis by showing representations trained with behavior cloning cluster around task-relevant states.

In this paper, we study visual representations for control through the lens of their capacity to recover the full environment state from images. Inspired by recent advances in structured visual representation learning for robotics (Qi et al., 2025; Radosavovic et al., 2022; Zhang et al., 2025b), we hypothesize that a representation's capacity to decode the underlying environment state — including its geometry, object structure, and physical attributes — is a strong indicator of its utility for downstream control. However, obtaining such state labels for real scenes is a major challenge. Instead, we capitalize on simulation environments (Todorov et al., 2012; Yu et al., 2020), which automatically provide access to the full ground truth state of the world (see Figure 1, left).

Specifically, we propose a uniform and compact representation of arbitrary environment states together with a lightweight state prediction head that can be applied to any visual backbone in Section 3. We select nine pre-trained representations, including both those specifically designed for robotics and general purpose ones (see Section 4 for details), for environment state regression and for policy learning and measure the correlation between the two tasks. Our study includes three simulation environments: general-purpose MetaWorld (Yu et al., 2020), RoboCasa (Nasiriany et al., 2024), which features a significant distribution shift between train and test samples, as well as the environment designed by Li et al. (2024) specifically to match the real world.

Across all the environments, state prediction accuracy demonstrates strong correlation with the success rate of the policies, as shown in Figure 3. It significantly outperforms all existing proxy metrics (see Figure 1), including the ones proposed by Burns et al. (2024), generalizes better across learning settings, and is substantially less computationally demanding (see Section 5). Crucially, we demonstrate that conclusions drawn from simulation reliably transfer to analogous real-world tasks, establishing our proxy as a practical tool for representation selection. Finally, our analysis reveals that representational demands vary across environments, and points to learning to encode the full state of the world as a promising direction for improving visual representations in control.

## 2 RELATED WORK

**Representation learning for manipulation.** Early methods learned to encode the image end-to-end with the policy via self-supervised or contrastive objectives applied to robot observations. CURL (Laskin et al., 2020) and Act (Zhang et al., 2022) leverage contrastive losses to align visual features with control signals, demonstrating improved sample efficiency in imitation and reinforcement learning. Scaling up to Internet-scale egocentric video, R3M (Nair et al., 2022) trains on diverse human first-person footage to distill a frozen backbone that transfers effectively to robotic imitation tasks. Masked reconstruction paradigms like MVP (Radosavovic et al., 2022) and RPT (Radosavovic et al., 2023) further show that reconstructing masked patches of robot videos—augmented with proprioceptive cues—yields task-agnostic representations that generalize across simulators and real platforms. Recently, VC-1 benchmarks dozens of vision backbones on a suite of simulated manipulation tasks, revealing that no single off-the-shelf model dominates across all challenges (Majumdar et al., 2023a). Most recently, (Burns et al., 2024; Gupta et al., 2024) demonstrate that Internet-scale pretraining often outperforms robotics-specific objectives in out-of-distribution evaluations.

**Robust visual representation learning.** In recent years, large-scale visual representation learning approaches have shown great robustness and generalizability in the wild (Akbari et al., 2021; Assran et al., 2023; Bao et al., 2022; Bardes et al., 2023; 2024; Chen & He, 2021; Gidaris et al., 2018; Girdhar et al., 2023; Grill et al., 2020; Joulin et al., 2016; LeCun, 2022; Li et al., 2022a; Mahajan et al., 2018; Pathak et al., 2016; Tong et al., 2022; Wang et al., 2023b; 2022; Xu et al., 2021; Zellers et al., 2022; Zhang et al., 2016; Zhou et al., 2022). Supervised ResNets set the early standard for feature transfer (He et al., 2016), followed by contrastive self-supervision (Chen et al., 2020; He et al., 2020)) and masked auto-encoder (He et al., 2022b). The Vision Transformer (ViT) introduces patch-based tokenization for scalable transformer encoders (Dosovitskiy et al., 2021), and subsequent self-distillation in DINOv2 yields even more generalizable features (Oquab et al., 2024). Multimodal alignment also demonstrates strong zero-shot capabilities by grounding images in natural language (Li et al., 2022a;b; 2023; Radford et al., 2021). Most recently, Internet-scale generative models for image (Peebles & Xie, 2023; Rombach et al., 2022) and video (Blattmann et al., 2023; Wang et al., 2023a) domains have emerged as powerful representations of the visual world.

While these representations excel in broad-domain tasks, systematic studies of their selection for manipulation remain scant. Preliminary work examines domain shifts between Internet images and egocentric robot views (Zhan et al., 2023), but lacks actionable guidelines for choosing among dozens of models. Burns et al. (2024) propose to use segmentation ability of a model as a proxy for its downstream manipulation performance, but their findings have been limited in scope and generality. Our state prediction objective addresses this gap by providing an efficient ranking of pre-trained visual representations that shows strong correlation with downstream policy performance across a wide variety of backbones, environments and learning settings.

**Analysis of visual representation for control.** A parallel line of work studies why certain visual encoders help manipulation. Qi et al. (2025) uncovers that visual representation naturally cluster around task relevant environment states when trained with behavior cloning objectives. Similar insights occur in (Jiang et al., 2025) which presents "manipulation-centric" analyzes regarding how much an embedding attends to object-contact regions. Multitask architectures like PerAct further highlight the benefit of strong visual features for manipulation tasks (Shridhar et al., 2023). Our work extends this line of research by providing a principled, state-space–level analysis of visual representations and showing which aspects of environment state drive manipulation performance.

**Evaluation of policy learning approaches.** Policy-learning benchmarks span both real-world and simulation environments. Real-world benchmarks like BridgeData V2 (Walke et al., 2023), RT-1/2 (Brohan et al., 2022a; 2023) and DROID (Khazatsky et al., 2024) collect thousands of demonstration trajectories across dozens of tabletop scenes on few manipulator embodiments, offering a public yardstick for behavior cloning and reinforcement learning (RL). However, real-world data collection remains expensive and evaluations slow and challenging to reproduce. To address these limitations, simulation suites have seen great improvement in recent years, providing rapid and reproducible comparison at scale, making them the de-facto testbeds for algorithmic research (Makoviychuk et al., 2021; Yu et al., 2020; Zhang et al., 2025a; Zhu et al., 2020).

Encouragingly, evidence suggests that carefully designed simulators can replace costly real-world trials during early development: SimplerEnv recreates RT-1-family policies in MuJoCo (Todorov et al., 2012) and recovers their relative ranking across camera and robot variants, validating sim-to-real evaluation as a faithful proxy (Li et al., 2024). Our study builds on these insights, and further capitalizes on the fact that simulation environments provide automatics access to the full state of the environment, enabling direct evaluation of the environment-state information encoded in pretrained visual representations. This approach allows visual representations for control to be selected without expensive policy rollouts, either in the real world or in simulation.

## 3 WORLD STATE ENCODING AND PREDICTION

As a *proxy task* for measuring the quality of visual backbones in policy learning, we regress a compact, simulator-grounded state of the world from raw images, shown in Figure 2. High accuracy on this task indicates that the backbone has learned the geometric, material, and kinematic cues needed for manipulation. Below, we first describe our compact, universal state representation format and then detail the design of our state prediction head and provide the corresponding losses.

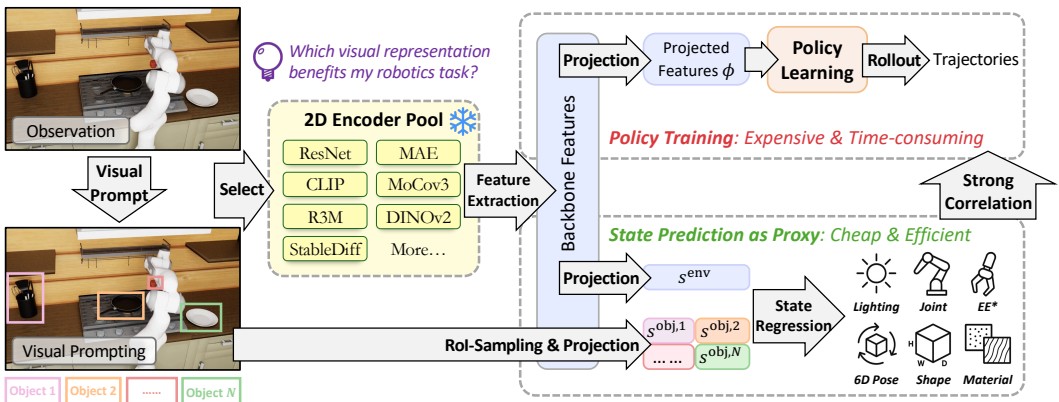

Figure 2: Our framework for efficient visual representation selection for control. We capitalize on the availability of ground truth world state information in simulators and propose a universal, compact encoding of the states, together with a light-weight state prediction head (bottom). We demonstrate a strong correlation between our proxy objective and downstream policy performance (top).

**Unified Low Dimensional States.** Low-dimensional state information captures the true configuration of the environment with high fidelity: exact robot kinematics, object geometry, material type, and lighting parameters without perceptual noise. We adopt a single, unified state representation that applies across all environments, so that regression error directly reflects visual representation's capacity in a task-agnostic way. We strive to capture all the state information that is provided by modern simulators, to allow for an out-of-the-box applicability to new environments in the future. Specifically, our state representation comprises $N_o + 1$ vectors: one *environment-level* vector for global states, and $N_o$ *object-level* vectors for each object (shown in the lower part of Figure 2). This decomposition allows us to analyze per-object details (pose, shape, material) and overall scene and agent configuration (lighting, joint state, end-effector pose) in isolation.

For each object $i = 1, \dots, N_o$ in the scene, the object-level vector is defined as

$$s^{\text{obj},i} = \left[p_{\text{pose}}^i, q_{\text{pose}}^i, s_{\text{shape}}^i, m_{\text{mat}}^i\right] \in \mathbb{R}^{3+4+3+M}, \tag{1}$$

where $p_{\text{pose}}^i \in \mathbb{R}^3$ represents position and $q_{\text{pose}}^i \in \mathbb{R}^4$ orientation (quaternion) of the object, $s_{\text{shape}}^i \in \mathbb{R}^3$ represent bounding-box of the object, and $m_{\text{mat}}^i \in \{0, 1\}^M$ is a one-hot vector over $M$ materials.

The single scene-level vector is defined as

$$s^{\text{env}} = \left[\ell, \ q^J, \ p^{\text{ee}}\right] \in \mathbb{R}^{1+N_j+N^{ee}}, \tag{2}$$

where $\ell \in \mathbb{N}$ are lighting categories, $q^J \in \mathbb{R}^{N_j}$ are robot joint angles, and $p^{\text{ee}} \in \mathbb{R}^{N_{ee}}$ is end-effector pose. Concatenation of these vectors yields a compact representation of the entire environment state

$$s = \left[s^{\text{obj},1}, \dots, s^{\text{obj},N_o}, \ s^{\text{env}}\right] \in \mathbb{R}^D, \quad D = N_o(3 + 4 + 3 + M) + (1 + N_j + N_{ee}). \tag{3}$$

Predicting these $N_o + 1$ vectors from images provides a clear and quantitative metric of the backbone's ability to encode both object-centric and scene-wide information. The details of the ground truth extraction for all state vectors from simulators are provided in the appendix.

**Visual-Prompted State Prediction.** To provide an efficient proxy metrics for representation selection, our approach outputs all the state vector dimensions in a single forward pass. However, predicting a *specific* object's state in a multi-object scene is inherently ambiguous unless the network is told where to look. Therefore, we design a visual-prompted state prediction setting, where 2D bounding boxes of target objects are provided to the model, so that the backbone can focus on relevant image regions.

*RoI pooling on the backbone feature map.* Given an input image $x$ and its corresponding feature map $\phi(x) \in \mathbb{R}^{H \times W \times C}$, we perform a region-of-interest (RoI) average pooling inside each object bounding box $b^i = [x_1, y_1, x_2, y_2]$:

$$u^i = \frac{1}{|b^i|} \sum_{(u,v) \in b^i} \phi(x)_{u,v} \in \mathbb{R}^C, \tag{4}$$

and map $u^i$ to $s^{\text{obj},i}$ via a single linear layer.

*Global encoding for environment-level factors.* To capture scene-wide states (lighting, robot joints, and end-effector poses), we apply a global average pooling over all spatial locations via

$$v = \frac{1}{HW} \sum_{u=1}^{H} \sum_{v=1}^{W} \phi(x)_{u,v} \in \mathbb{R}^C. \tag{5}$$

Similarly, $v$ is projected to $s^{\text{env}}$ with a linear layer. Both projections layers are optimized against the simulator ground truth, masking out the objects that are not visible in the current frame.

**State Encoding & Loss Function.** Discrete states such as material type ($M$ classes), lighting preset ($L$ options), and object shape modeled as a 3D box with each edge quantized into $S$ bins are represented by one-hot vectors $\phi_k(c_k) \in \{0,1\}^{n_k}$. These vectors are predicted via a softmax layer and trained with a cross-entropy loss.

Continuous states, including object position $p_{\text{pose}}$ and rotation $q_{\text{pose}}$, robot joint states $N_J$ and end-effector pose $p^{ee}$ are standardized per dimension: $z_i = \frac{x_i - \mu_i}{\sigma_i}$, where $\mu_i$, $\sigma_i$ are empirical mean and standard deviation from training data. The network then regresses these normalized values using an $L_2$ loss. Continuous modeling is used for these states as they are closely related to action planning.

**Proxy Metric.** To ensure broad coverage, we include all the dimensions of the state vector $s$ in Equation 3. Specifically, for categorical states, we compute classification accuracy, while for continuously states, we use the negative mean squared error (MSE) as the score, so that higher is better for all metrics. After collecting scores for all models across all states, we apply min-max normalization per states across models. Finally, we compute the mean of the normalized scores across states to obtain a single evaluation score for each model. Let $\mathcal{A}$ denote the set of states and $\mathcal{M}$ the set of models. The final proxy score for model $m \in \mathcal{M}$ is defined as:

$$S_m = \frac{1}{|\mathcal{A}|} \sum_{a \in \mathcal{A}} \frac{r_{m,a} - \min_{\tilde{m} \in \mathcal{M}} r_{\tilde{m},a}}{\max_{\tilde{m} \in \mathcal{M}} r_{\tilde{m},a} - \min_{\tilde{m} \in \mathcal{M}} r_{\tilde{m},a}}, \tag{6}$$

where $r_{m,a}$ is the raw score of model $m$ on state $a$.

## 4 EXPERIMENT PROTOCOL

In this work, we set out to analyze what makes visual representations effective for generalist manipulation policies. Specifically, we evaluate the correlation between scores of various candidate objectives and the success rates of policies learned on top of a wide variety of pre-trained visual representations. We validate our findings across three distinct simulation environments and further test whether our conclusions transfer to the real world, where ground-truth state labels are unavailable. To this end, we show that state prediction accuracy measured in simulation reliably predicts backbone performance on real-world tasks. Next, we describe the details of our experiential setup.

**Simulation environments.** We evaluate our approach in three distinct simulation environments: MetaWorld 50 (Yu et al., 2020), RoboCasa (Nasiriany et al., 2024), and SimplerEnv (Li et al., 2024). *MetaWorld 50* is a widely used benchmark for multitask robotic manipulation, offering a diverse suite of 50 object-centric challenges that rigorously test both generalization and dexterity. *RoboCasa* (Nasiriany et al., 2024) offers realistic household environments with everyday objects and furniture, consisting 24 atomic tasks. *SimplerEnv* is a lightweight and highly customizable platform that focuses on closely matching the simulation with real world setups from Google RT-1/2 (Brohan et al., 2022a; 2023) and BridgeV2 (Walke et al., 2023). We adopt 10 different tasks from it, including 6 tasks for Google robots and 4 tasks for WidowX. Evaluating across diverse simulation domains helps ensure generalization over object categories, task complexity, and dynamics, reducing overfitting to any one setting. For all environments, we train the models with 50 demonstrations per task. In Metaworld (Yu et al., 2020), we generate demonstrations using the provided expert policy. For RoboCasa, we use the official set of 50 human demonstrations per task. In SimplerEnv, we collect demonstrations using our own expert policy. The success rates are reported with 100 rollouts per task.

**Real-world environments.** To validate the sim-to-real transferability of our proxy, we reproduced two tasks from the WidowX benchmark (Li et al., 2024) on a physical Xarm6 robot arm (UFACTORY) equipped with the UFactory Xarm Gripper. A RealSense D455 camera (Intel RealSense) was mounted

in the corner to provide visual input. The control frequency was set to 10 Hz with a maximum end-effector speed of 0.1 m/s. Specifically, we reproduced the `widowx_carrot_on_plate` and `widowx_stack_cub` tasks and report their average success rate. To better match the original WidowX environment, we adjusted the camera pose and used a similar table covering (see Figure 6, left). We collected 100 demonstrations per task and report success rates over 100 rollouts.

**Pretrained models.** We include the following pretrained visual representations in our evaluation. Detailed description of checkpoint, training data, and implementation is in the Appendix Section D.

*Supervised models.* We evaluate conventional ImageNet–trained backbones, including ResNet-18 (He et al., 2016) and Vision Transformer (ViT-B) (Dosovitskiy et al., 2021), as well as the vision encoder from CLIP model (Radford et al., 2021), which couples the same ViT architecture with a contrastive language–image training objective.

*Self-supervised models.* For label-free representation learning we cover MoCo-v3 (momentum contrast for ViTs) (He et al., 2020), MAE (masked auto-encoder) (He et al., 2022b), DINO v1/v2 (self-distillation without labels) (Caron et al., 2021; Oquab et al., 2024). All three are trained on Internet images and serve as domain-agnostic baselines without robotics-specific biases.

*Manipulation-specific models.* To probe representations explicitly optimized for robotic control, we separately consider R3M (Nair et al., 2022), which learns reward-predictive features from egocentric manipulation videos. Evaluating this encoder allows us to quantify the added value of task-aligned pre-training relative to more generic visual representations.

*Generative models.* Finally, we explore the potential of generative models that have demonstrated remarkable capabilities in visual synthesis. Stable Diffusion (SD) (Rombach et al., 2022) learns to generate images from noise guided by text prompts, modeling diverse visual scenes that may provide useful physical priors.

**Proxy objectives.** We group the commonly evaluated proxy objectives into two categories: environment-agnostic tasks and environment-specific tasks. The environment-agnostic ones include: *ImageNet Recognition* – Linear probing on ImageNet (Deng et al., 2009), and *Shape Bias* – Measuring shape bias on Stylized-ImageNet (Geirhos et al., 2018). The environment-specific tasks include: *Few-Shot Learning* – five demonstrations per task for efficient training and evaluation (Burns et al., 2024); *Action MSE* – action prediction mean squared error loss; *Segmentation* – Semantic segmentation performance (Burns et al., 2024; Man et al., 2024); *Depth Estimation* – Monocular depth estimation (Banani et al., 2024; Burns et al., 2024). A thorough description of these baseline proxies are detailed in the Appendix Section B.

**Policy learning.** We adopt a multi-task diffusion policy conditioned on text prompts for policy learning from the original design (Chi et al., 2023; Ze et al., 2024), but also evaluate robustness of our findings to different behavior cloning algorithms in Section A.2 in the appendix. A CLIP text encoder (Radford et al., 2021) embeds each instruction and fuses it with visual features. Action is generated via standard diffusion forward/reverse passes with classifier-free guidance, enabling diverse task handling without retraining task-specific heads. The experiments in the main paper primarily utilize frozen visual representations; however, we also report correlation results with finetuned backbones for completeness. Detailed training setup and implementation are provided in Section E.

**Evaluation protocol.** To provide an objective way of assessing whether a proxy can reliably stand in for expensive policy training when comparing visual encoders, our evaluation protocol closely follows the SimplerEnv benchmark (Li et al., 2024). We use the Mean Maximum Rank Violation (MMRV) (Li et al., 2024) metric to measure the ranking correlation between the policy success rate $R_i$ and the state prediction proxy scores $S_i$ of a set of visual models. Specifically, we first compute the pairwise violation

$$\text{RankViolation}_{ij} = \left| R_i - R_j \right| \mathbf{1}\left[ (S_i < S_j) \neq (R_i < R_j) \right], \quad (7)$$

where $i, j \in \mathcal{M}$ are different visual models. Then, we aggregate each policy's worst-case error:

$$\text{MMRV} = \frac{1}{N} \sum_{i=1}^{N} \max_{1 \leq j \leq N} \text{RankViolation}_{ij}. \quad (8)$$

In addition, we report the complementary Pearson's correlation (Kadian et al., 2020; Pearson, 1895) metric $r(R, S)$ to capture overall linear agreement. Low MMRV and high $r$ indicate strong ranking fidelity.

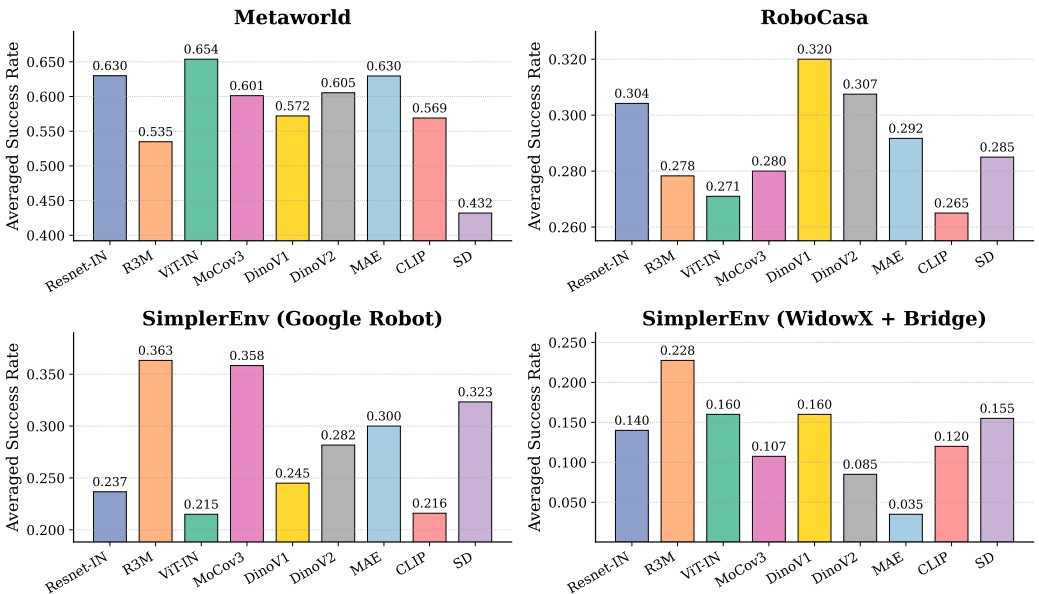

Figure 3: Evaluation of a diverse set of visual backbones in different robotic simulation environments. From top left to bottom right: The visually simple Metaworld (a) favors more traditional ImageNet-pretrained representations. RoboCasa (b) requires precise objet localization and thus favors with strong object priors, whereas realistic SimplerEnv environments (c, d) benefit from real world robot data pre-training.

## 5 EXPERIMENTAL RESULTS

**Assessment of pre-trained models.** Pre-trained visual representations promise to boost policy performance in robotics, where on-policy training data is scarce. Yet in practice their impact is influenced by factors such as domain shift, network architecture, and training setup. In this section, we present a systematic comparison of diverse pretrained vision encoders across multiple benchmarks, asking whether their relative performance rankings are consistent or inherently environment-dependent.

In Figure 3, we evaluate visual backbones across four distinct domains. In Metaworld, which features non-photorealistic observations, large-scale pretrained models such as CLIP and DINOv2 underperform. This is likely due to a significant distribution shift between high-resolution Internet images used in training these models and cartoonish frames in Metaworld. In contrast, ImageNet-pretrained models exhibit more stable performance. In RoboCasa, the combination of high-fidelity rendering and diverse scene layouts favors models like DINOv1 and DINOv2, which are particularly effective at object localization. In SimplerEnv, which is designed to mimic real-world indoor scenes, R3M achieves strong performance, benefiting from its pretraining on real robot observations. These experiments show that *there is no single optimal representation* for robot manipulation. Thus, exploring which factors influence the effectiveness of visual representations is an important problem.

**Proxy evaluation in simulation.** We now evaluate of state regression objective as a proxy for predicting the success rate of downstream policies. To this end, we first plot the correlation between the proxy score and the success rate of the corresponding policy in Figure 4. We observe that, across all 4 environments, the success in predicting the unified environment state representation proposed in Section 3 exhibits strong correlation with the effectiveness of the polices, as indicated by low MMRV and high Pearson correlation ($r$) scores. The mistakes largely come from some backbones achieving virtually equal success rates (e.g. for ViT, DINOv1 and StableDiffusion in the bottom right plot), making differentiation extremely challenging.

To put these results in context, we compare against several proxy objectives and approaches for representation selection proposed in the past in Table 1, including some requiring full policy evaluation (Few-Shot) and full policy training (Action MSE). Remarkably, our policy-free approach shows stronger correlation with the success rate of the policy compared even to these privileged baselines in all the environments. Among the purely visual proxies, the segmentation metric proposed in (Burns et al., 2024) shows strong performance on the RoboCasa benchmark, which requires accurate object localization, but significantly underperforms in all the other environments. This observation validates our claim that focusing on individual aspects of visual reasoning limits generalization.

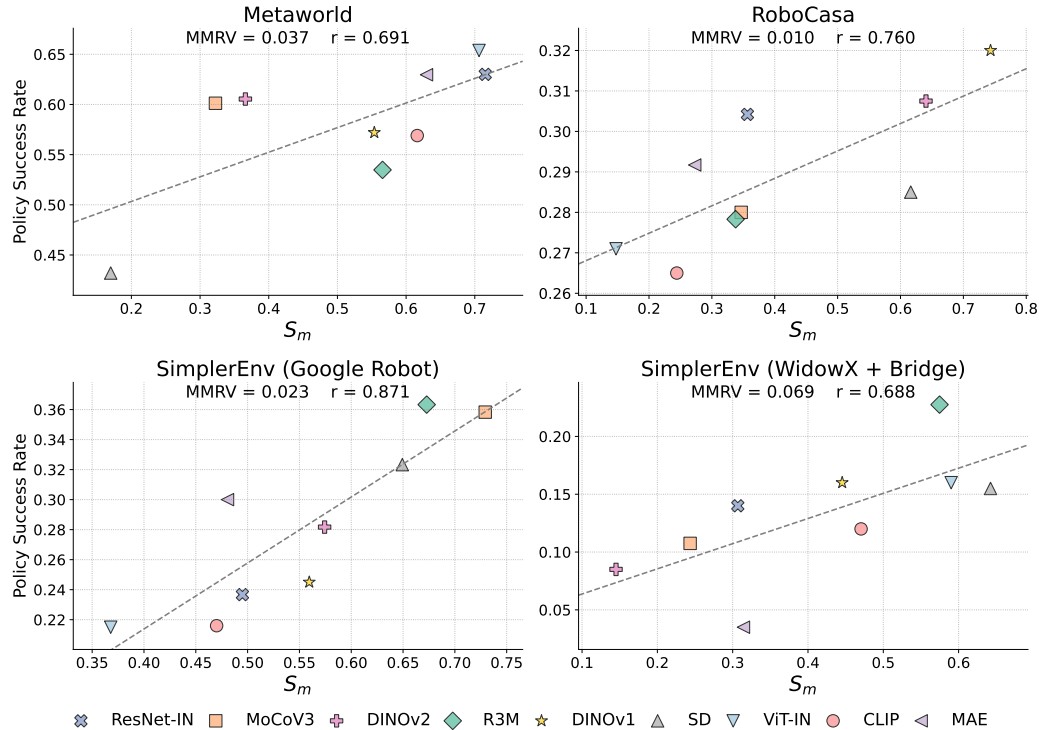

Figure 4: **Correlation between state prediction score and success rate of the policies.** Our proposed proxy task shows a strong correlation and MMRV score in all four different environments.

| | Few-Shot | Action MSE | ImageNet | Shape Bias | Segmentation | Depth | Ours |
|---|---|---|---|---|---|---|---|
| | | | **MMRV** $\downarrow$ | | | | |
| Metaworld | 0.069 | 0.081 | 0.070 | 0.175 | 0.167 | 0.109 | **0.037** |
| RoboCasa | 0.028 | 0.041 | 0.245 | 0.041 | 0.023 | 0.035 | **0.010** |
| SimplerEnv (G) | 0.060 | 0.108 | 0.123 | 0.066 | 0.115 | 0.101 | **0.023** |
| SimplerEnv (W) | 0.113 | 0.124 | 0.124 | 0.090 | 0.113 | 0.137 | **0.069** |
| Average | 0.068 | 0.089 | 0.141 | 0.093 | 0.105 | 0.096 | **0.035** |
| | | | **Pearson** $r$ $\uparrow$ | | | | |
| Metaworld | 0.650 | 0.016 | 0.587 | -0.905 | -0.268 | 0.089 | **0.691** |
| RoboCasa | 0.243 | -0.340 | 0.242 | -0.291 | 0.527 | 0.083 | **0.760** |
| SimplerEnv (G) | 0.624 | -0.465 | -0.660 | 0.633 | -0.475 | 0.027 | **0.871** |
| SimplerEnv (W) | -0.128 | -0.385 | -0.250 | 0.286 | 0.048 | -0.483 | **0.688** |
| Average | 0.347 | -0.294 | -0.020 | -0.069 | -0.042 | -0.071 | **0.753** |

Table 1: Comparison of visual backbone selection proxies on four different simulation environments using MMRV and Pearson correlation. Our state prediction objectives show top performance across all the environments, outperforming even the methods that have direct access to policy (Few-Shot and Action MSE).

Finally, although freezing the visual backbone is more resource-efficient, fine-tuning is a more typical approach in robot learning. To analyze the full potential of various backbones, we fine-tuned them in MetaWorld (Yu et al., 2020) and report the results in Figure 5. Our method still achieves high correlation for fine-tuned models. More details are provided in Appendix Section A.1

**Proxy evaluation in the real world.** Due to the high cost of policy evaluation in the real world, we only report results with a representative subset of the visual backbones used in the simulation experiments (ResNet-IN (He et al., 2016), R3M (Nair et al., 2022), VIT-IN (Dosovitskiy et al., 2021), CLIP (Radford et al., 2021) and MAE (He et al., 2022b)). Figure 6 (right) shows the correlation between their *real-world* success rates and state prediction accuracy in the *simulated* version of WidowX. We find a strong correlation, with MMRV and $r$ scores comparable to those observed in simulation-only experiments (Figure 4, bottom right), despite the significant domain gap.

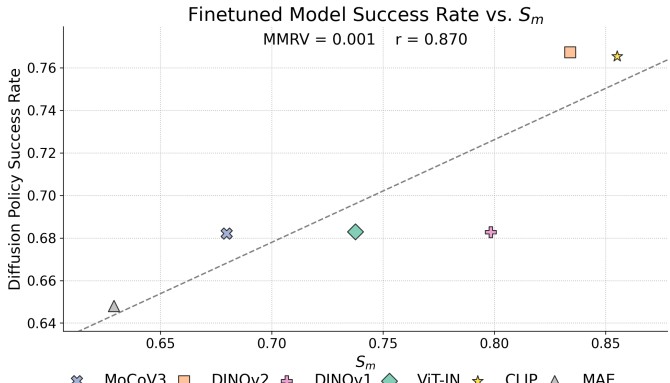

Figure 5: **Correlation for fine-tuned backbones in MetaWorld.** Our proxy demonstrates strong correlation with policy performance when visual backbones are fine-tuned with the policy objective.

| | $p_{\text{pose}}$ | $q_{\text{pose}}$ | $s_{\text{shape}}$ | $m_{\text{mat}}$ | $q^J$ | $p^{ee}$ | $l$ | Full ($S_m$) |
|---|---|---|---|---|---|---|---|---|
| Metaworld | 0.055 | 0.089 | **0.032** | 0.153 | 0.038 | 0.069 | 0.073 | 0.037 |
| RoboCasa | 0.014 | 0.017 | **0.011** | 0.024 | 0.032 | 0.027 | 0.031 | 0.010 |
| SimplerEnv (G) | 0.045 | 0.100 | 0.082 | 0.120 | 0.075 | **0.038** | 0.084 | 0.023 |
| SimplerEnv (W) | 0.126 | **0.032** | 0.126 | 0.107 | 0.048 | 0.048 | 0.095 | 0.069 |

Table 2: **Predictive power of individual state dimension.** We report MMRV for individual state components (as defined in Section 3) across the 4 simulation environments. They place different demands on representations, but our full approach provides a versatile proxy for backbone selection.

These results demonstrate that simulation-based state prediction is an effective proxy for real-world policy performance, reinforcing our central message that the capacity to capture state is a defining characteristic of visual representations that generalize well to control.

**Analysis of environment-specific demands.** Our analysis above shows that the capacity of a backbone to encode the *full* state of the world is an *environment-agnostic* proxy for its utility for policy learning. However, from Figure 3 we can also observe that different environments present different representational requirements. Here we study whether dissecting the effect of the individual state components can illuminate the unique demands of each domain. To this end, we conduct a fine-grained analysis of the full state vector by measuring the predictive power of each individual subset of dimension in Table 2. We observe that different environments indeed present different demands for visual representations, with Metaworld and RoboCasa requiring accurate object localization in 2D ($s_{shape}$) and SimplerEnv environments benefiting the most from accurate 3D end effector pose ($p^{ee}$) and 3D object orientation estimation ($q_{pose}$). Notably, while our approach allows to obtain insights into the individual demands of each manipulation environment, simply regressing the entire state

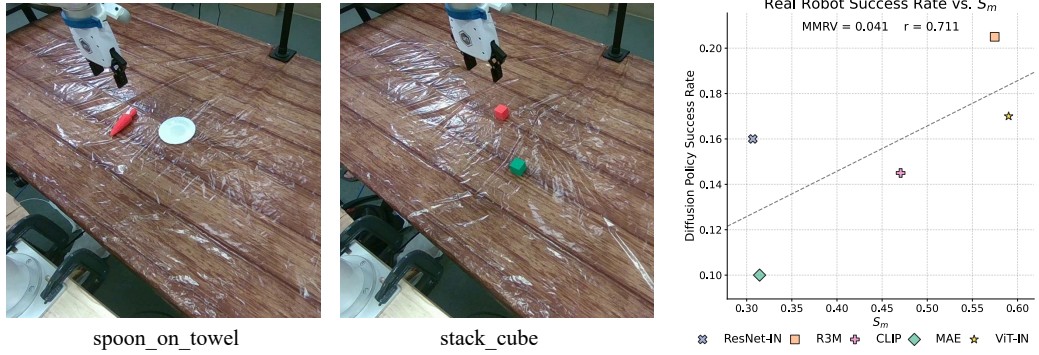

Figure 6: **Real-world experiment setting and results.** State prediction accuracy in the *simulated* WidowX environment strongly correlates with real-world success rates (averaged over two tasks), supporting the practical applicability of our proxy and reinforcing the link between state representations and control performance.

|  | ViT-IN | Mocov3 | MAE | CLIP | DINOv2 |
|---|---|---|---|---|---|
| Baseline | 0.683 | 0.671 | 0.648 | 0.765 | 0.767 |
| State prediction training | **0.740** | **0.743** | **0.712** | **0.801** | **0.795** |

Table 3: **State prediction for representation learning.** Joint fine-tuning with state prediction consistently improves downstream policy success rates across multiple visual backbones on MetaWorld.

serves as an effective, versatile proxy, as indicated by the strong performance of our full method in the right most column. We provide further analysis of the effect of individual state dimensions on our method's performance in Section A.6 in the Appendix.

**State prediction as representation learning.** Beyond its utility as a proxy for model selection, fine-tuning vision encoders with our state prediction objective delivers substantial gains. We validate this by jointly fine-tuning five visual backbones with both policy learning and state regression on MetaWorld (Yu et al., 2020). Specifically, we conduct experiments on ViT-IN (Dosovitskiy et al., 2021), MocoV3 (He et al., 2020), MAE (He et al., 2022b), CLIP (Radford et al., 2021) and DINOv2 (Oquab et al., 2024), using our objective defined in Section 3 alongside the diffusion policy training loss.

During joint training, we attach a state-prediction head to the visual backbone and optimize a joint loss:

$$\mathcal{L}_{\text{joint}} = \mathcal{L}_{\text{policy}} + \lambda \mathcal{L}_{\text{state}}, \tag{9}$$

where $\mathcal{L}_{\text{policy}}$ is the diffusion policy loss and $\mathcal{L}_{\text{state}}$ is the regression loss. Gradients from both terms are backpropagated through the encoder, while the policy and state heads are updated only by their respective losses. This setup encourages the backbone to retain both state-relevant information and action prediction ability.

As shown in Table 3, explicitly predicting environment state improves downstream policy performance, pointing toward predictive world modeling as a promising direction for advancing visual representation learning in robotics. We provide further analysis of our auxiliary objective in Section A.7 in the Appendix.

**Computational complexity.** Finally, we evaluate the total computational cost (training and evaluation) of each policy learning and proxy task, using a single A100 GPU. The results achieved in Metaworld are reported in Table 4, and demonstrate that our state prediction is not only more effective as a proxy for the visual representation selection, but also more computationally efficient than all other alternatives. The comparison of results between the ResNet and ViT models also demonstrates that the computational advantages are more significant for larger models. Further details are provided in the Appendix Section C.

|  | Few-Shot | Action MSE | ImageNet | Shape Bias | Segmentation | Depth | Ours |
|---|---|---|---|---|---|---|---|
| ResNet-IN | 78 min | 45 min | 11 min | 23 min | 10 min | 10 min | **4 min** |
| ViT-IN | 113 min | 72 min | 46 min | 92 min | 27 min | 27 min | **12 min** |

Table 4: **Analysis of computational efficiency.** We report the time for joint training and evaluation of all representation selection approaches (As defined in Section 4). Our state prediction proxy objective is not only more effective in reflecting policy learning performance, but also more computationally efficient than all the alternatives. Results are computed with a single A100 GPU.

## 6 CONCLUSION

In this work, we analyze what makes visual representations useful for control by leveraging simulation environments with access to ground-truth state. Using state prediction error as a proxy, we showed that representations capacity of capturing environment state correlates much more strongly with downstream policy success than existing baselines. We have also demonstrated that our conclusions hold in the real world. These results suggest a simple rule of thumb for visual representation learning in robotics: the better a model understands the world, the better it can act in it.

## ACKNOWLEDGMENTS

This work was supported in part by the Toyota Research Institute, NSF under Grants 2106825 and 2519216, the DARPA Young Faculty Award, the ONR Grant N00014-26-1-2099, and the NIFA Award 2020-67021-32799. This work used computational resources, including the NCSA Delta and DeltaAI supercomputers through allocations CIS230012, CIS230013, CIS240387, and CIS250829 from the Advanced Cyberinfrastructure Coordination Ecosystem: Services & Support (ACCESS) program, as well as the TACC Frontera supercomputer, Amazon Web Services (AWS), and OpenAI API through the National Artificial Intelligence Research Resource (NAIRR) Pilot.

## REPRODUCIBILITY STATEMENT

We have made extensive efforts to ensure the reproducibility of our work. Detailed descriptions of our experimental setup, including the unified environment state representation, state prediction objective, and evaluation metrics (MMRV and Pearson correlation), are provided in Section 3 and Section 4 of the main paper. Full implementation details, including pretrained model checkpoints, training hyperparameters, and proxy objectives, are documented in Appendix Sections D and E. We also provide a comprehensive analysis of additional baselines, robustness across policy algorithms, and computational complexity in Appendix Sections A–C. For simulation environments, we describe how demonstrations and ground-truth states are collected in Appendix Section D, while real-world experiments are specified in Section 4.2, including robot hardware, control frequency, and demonstration protocol. Together, these materials ensure that all reported results can be replicated and extended by the community.

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

In this appendix, we provide additional experiments, implementation details, and visualizations that could not be included in the main paper due to space constraints. We begin by reporting further details for the fine-tuning experiment in Figure 5 in the main paper in Section A.1. In addition, in Section A.2, we evaluate our approach using an alternative behavior cloning algorithm to assess robustness across policy algorithms. In Section A.3, we show more results for the few-shot baseline and Section A.4 reports error bar analysis and statistical robustness of our model. Furthermore, Section A.5 explores the transferability of our proxy to RoboCasa navigation tasks, and Section A.6 provides an analysis of different simulator state subsets. Section A.7 details further experiments on joint learning, Section A.8 evaluates additional robotics-specific encoders, and Section A.9 analyzes our method's reliance on simulation versus real-world states. We provide additional discussion regarding DINOv1 and v2 models in Section A.10 and show failure cases in Section A.11. We detail the settings of other commonly used proxies in Section B and report their latency analysis in Section C. We then present details of the simulation environments used in our experiments in Section D, including how we obtain state information, determine object visibility, and collect expert demonstrations. Section E provides additional implementation details, Section F discusses the limitation and broader impact of our work, and Section H report per-task policy evaluation results for all the experiments in Figure 3 in the paper.

# A  ADDITIONAL RESULTS AND ANALYSIS

## A.1  DETAILS OF FINETUNING RESULTS

We notice that fine-tuning different backbones requires different hyperparameters. Thus, we conduct experiments on the most common ViT backbones in our setting. In the experiments, the learning rate of the backbone is set to $1 \times 10^{-6}$ to stabilize the training. For the state regression proxy task, the backbone is set to the same $1 \times 10^{-6}$ learning rate. Additionally, we observe that all the backbones' performance improves compared with frozen parameters, but with heavily unequal improvements (DINOv2 improves a lot while MoCov3 improves minimally). Such results demonstrate that the performance ranking tends to change during backbone finetuning.

## A.2  ROBUSTNESS ACROSS POLICY ALGORITHMS

To further validate our approach, we change the diffusion policy to the MLP-based policy algorithm (Nair et al., 2022). Specifically, we use three layers to predict the action. As shown in Figure I, the algorithm change doesn't affect the performance correlation significantly, demonstrating the robustness of our proxy objective. Overall, the MLP prediction performs somewhat worse than the diffusion policy.

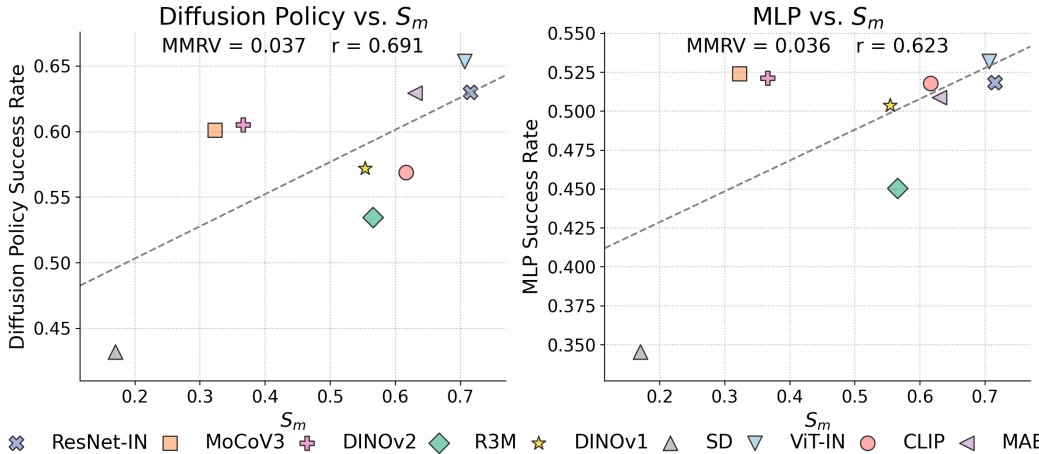

Figure I: The performance correlation still holds when changing the behavior cloning algorithm to MLP algorithm, demonstrating our method's robustness.

A.3    FURTHER RESULTS FOR THE FEW-SHOT BASELINE

In the paper, the few-shot proxy is evaluated over five episodes for efficiency, which limits its predictive power. Here we further evaluate the few-shot learning proxy using the entire evaluation set. Specifically, we take 100 episodes from the MetaWorld and SimplerEnv environments and 50 episodes from the RoboCasa environment using their default settings. As the results in Table I show, this few-shot learning variant exhibits the best correlation among all the baselines on the simplistic Metaworld benchmark. However, in more challenging RoboCasa and SimplerEnv, this proxy still struggles to accurately capture the policy learning performance of different visual representations. Crucially, it is about $150\times$ more expensive than our proposed approach, making it an impractical choice even for relatively simple environments like Metaworld. For reference, full training and evaluation in MetaWorld with a ResNet backbone takes about 30 hours. In other words, our proxy reduces the computational cost by 99.8%.

| | MetaWorld | RoboCasa | SimplerEnv (G) | SimplerEnv (W) | Runtime |
|---|---|---|---|---|---|
| **MMRV $\downarrow$** | | | | | |
| Few-Shot (5 Episodes) | 0.069 | 0.028 | 0.060 | 0.113 | 78 min |
| Few-Shot (Full Episodes) | 0.042 | 0.037 | 0.097 | 0.104 | $\sim$600 min |
| Ours | **0.037** | **0.010** | **0.023** | **0.069** | 4 min |
| **Pearson $r \uparrow$** | | | | | |
| Few-Shot (5 Episodes) | 0.650 | 0.243 | 0.624 | -0.128 | 78 min |
| Few-Shot (Full Episodes) | **0.885** | -0.043 | 0.348 | 0.137 | $\sim$600 min |
| Ours | 0.691 | **0.760** | **0.871** | **0.688** | 4 min |

Table I: Full episode evaluation on the proxy task of few-shot learning. Our proxy performs better in most environments. The reported time is measured on MetaWorld with a ResNet backbone.

A.4    STATISTICAL ANALYSIS

We report error bars (mean $\pm$ Standard Error of the Mean, SEM) for policy performance across four environments. For each visual backbone, we conduct $n = 5$ independent training runs (different random seeds) and evaluate each trained policy using the same evaluation protocol as in the main paper. Across all environments, the SEM is generally much smaller than the performance gaps between methods, supporting the statistical robustness of our comparisons.

| Backbone | MetaWorld SR | RoboCasa SR | Simpler (G) SR | Simpler (W) SR |
|---|---|---|---|---|
| ResNet | 0.714 ($\pm$0.001) | 0.303 ($\pm$0.008) | 0.212 ($\pm$0.018) | 0.139 ($\pm$0.014) |
| ViT | 0.706 ($\pm$0.001) | 0.263 ($\pm$0.005) | 0.210 ($\pm$0.011) | 0.174 ($\pm$0.010) |
| MAE | 0.631 ($\pm$0.000) | 0.284 ($\pm$0.008) | 0.314 ($\pm$0.019) | 0.074 ($\pm$0.018) |
| CLIP | 0.616 ($\pm$0.001) | 0.261 ($\pm$0.006) | 0.240 ($\pm$0.015) | 0.115 ($\pm$0.011) |
| MoCov3 | 0.605 ($\pm$0.002) | 0.284 ($\pm$0.005) | 0.351 ($\pm$0.005) | 0.148 ($\pm$0.015) |
| DINOv2 | 0.598 ($\pm$0.005) | 0.317 ($\pm$0.005) | 0.250 ($\pm$0.011) | 0.090 ($\pm$0.005) |
| DINOv1 | 0.570 ($\pm$0.002) | 0.311 ($\pm$0.006) | 0.225 ($\pm$0.011) | 0.155 ($\pm$0.011) |
| R3M | 0.538 ($\pm$0.002) | 0.273 ($\pm$0.005) | 0.353 ($\pm$0.015) | 0.221 ($\pm$0.014) |
| SD | 0.438 ($\pm$0.007) | 0.282 ($\pm$0.005) | 0.325 ($\pm$0.015) | 0.163 ($\pm$0.017) |

Table II: Error bar analysis (mean $\pm$ SEM over $n = 5$ runs) across four environments. Values are reported to the thousandth.

In addition, we compute error bars for state regression in the MetaWorld environment using $n = 5$ independent runs and report the results below. We report the mean regression score $\pm$ Standard Error of the Mean (SEM) across runs. Consistent with the policy results, the SEM is small relative to the differences between encoders, supporting the robustness of the regression comparison.

| | ResNet | ViT | MAE | CLIP | MoCo v3 | DINOv2 | DINOv1 | R3M | SD |
|---|---|---|---|---|---|---|---|---|---|
| Regression Score | 0.715 (±0.001) | 0.706 (±0.002) | 0.630 (±0.001) | 0.616 (±0.002) | 0.323 (±0.003) | 0.366 (±0.002) | 0.554 (±0.001) | 0.566 (±0.003) | 0.170 (±0.005) |

Table III: **Error bars for state regression (MetaWorld).** We report mean regression score $\pm$ SEM over 5 runs for each encoder. All values are reported to the thousandth.

## A.5 TRANSFER TO ROBOCASA NAVIGATION TASKS

We also evaluate our proxy on RoboCasa *navigation* tasks, using exactly the same formulation as for manipulation. Despite the substantial difference in task objectives and success criteria, the proxy remains strongly correlated with navigation policy performance (Table IV), indicating that it generalizes beyond manipulation-only settings.

| | MMRV ↓ | Pearson's correlation ↑ |
|---|---|---|
| RoboCasa Manipulation (Ours) | 0.010 | 0.760 |
| RoboCasa Navigation (Ours) | 0.022 | 0.727 |
| RoboCasa Navigation (Segmentation) | 0.037 | 0.436 |
| RoboCasa Navigation (Depth) | 0.043 | 0.172 |

Table IV: **Proxy quality on RoboCasa manipulation vs. navigation.** The same proxy, applied without modification, achieves a strong correlation with navigation policy performance, demonstrating transfer across task families.

## A.6 ADDITIONAL ANALYSIS OF STATE SUBSETS

In this section, we further analyze the effect of different subsets of simulator state dimensions. We want to emphasize that the state dimensions are not hand-designed: our method uses all variables directly exposed by each simulator to define a uniform, task-agnostic state representation. Accordingly, the per-dimension and subset ablations below are not intended to show that every individual attribute contributes positively in isolation, but to illustrate how different environments emphasize distinct parts of the state and the effect of different subsets.

**Full state vs. task-critical subsets.** We first compare the full state to a "task-critical" subset consisting only of target object states and end-effector pose. As shown in Table V, task-critical subsets do not improve correlation. In MetaWorld they slightly reduce Pearson $r$, while in RoboCasa they mildly increase $r$ at the cost of worse MMRV. Overall, full states provide a more balanced and robust proxy.

| | Full state | | Task-critical subset | |
|---|---|---|---|---|
| Environment | MMRV ↓ | $r$ ↑ | MMRV ↓ | $r$ ↑ |
| MetaWorld | 0.037 | **0.691** | 0.037 | 0.615 |
| RoboCasa | **0.010** | 0.760 | 0.015 | **0.774** |

Table V: **Full state vs. task-critical subsets.** Using only task-critical variables does not consistently improve correlation, while full states remain competitive or better in terms of MMRV.

**Leave-one-out ablations.** To assess the contribution of individual state groups, we perform leave-one-out experiments over MetaWorld and RoboCasa. We remove one group at a time from the full state: object position (p_pose), object orientation (q_pose), shape (s_shape), material (m_mat), joint angles (q_j), end-effector pose (p_ee), and lighting (l). Results in Table VI show that dropping any group generally degrades performance relative to the full state; different environments are sensitive to different components, confirming that multiple state dimensions are important rather than a single hand-picked subset.

**Searching for best-performing subsets.** Finally, we enumerate combinations of state groups and select those that maximize Pearson correlation in each environment. As shown in Table VII, different

| Removed group | MetaWorld | | RoboCasa | |
|---|---|---|---|---|
| | $r \uparrow$ | MMRV $\downarrow$ | $r \uparrow$ | MMRV $\downarrow$ |
| p_pose | 0.7325 | 0.0402 | 0.7169 | 0.0163 |
| q_pose | 0.6358 | 0.0455 | 0.7342 | 0.0163 |
| s_shape | 0.7607 | 0.0361 | 0.7028 | 0.0186 |
| m_mat | 0.6979 | 0.0365 | 0.7797 | 0.0105 |
| q_j | 0.6153 | 0.0365 | 0.7744 | 0.0153 |
| p_ee | 0.6656 | **0.0308** | 0.7375 | 0.0134 |
| l | 0.6656 | 0.0365 | 0.7602 | **0.0099** |

Table VI: **Leave-one-out ablations over state groups.** Removing any state group generally harms either MMRV or Pearson $r$, and the most influential groups differ between environments, indicating that multiple components contribute to the proxy.

tasks favor different subsets (e.g., joint and orientation states in MetaWorld vs. position and shape in RoboCasa), and there is no universal best subset across environments. While tailored subsets can slightly improve metrics in a given setting, they require manual design and do not transfer. This supports our choice of using the full simulator state as a simple, task-agnostic representation that scales naturally with environment complexity.

| Environment | Best subset | MMRV $\downarrow$ | $r \uparrow$ |
|---|---|---|---|
| MetaWorld | q_pose, q_j | 0.0245 | 0.8702 |
| RoboCasa | p_pose, s_shape | 0.0110 | 0.8543 |

Table VII: **Best-performing state subsets (per environment).** Enumerating combinations of state groups reveals environment-specific optima but no universal subset, reinforcing the appeal of a single full-state formulation.

### A.7 ADDITIONAL EXPERIMENTS ON JOINT LEARNING

In addition to the main results in Table 3, we further provide additional experiments of using a state-prediction proxy during policy learning.

**Two-stage vs. joint training.** In addition to joint training, we explore the performance of two-stage training. Specifically, we first fine-tune the encoder with the regression objective and then fine-tune it again with the policy objective only. Table VIII shows that joint training consistently outperforms both direct fine-tuning and two-stage training across all backbones. While two-stage training improves over direct fine-tuning, it remains worse than joint training, suggesting that maintaining the auxiliary objective throughout optimization is more effective than pretraining the backbone and discarding the proxy signal.

| Method | ViT-IN | MoCoV3 | MAE | CLIP | DINOv2 |
|---|---|---|---|---|---|
| Direct fine-tuning | 0.683 | 0.671 | 0.648 | 0.765 | 0.767 |
| Joint training | **0.740** | **0.743** | **0.712** | **0.801** | **0.795** |
| Two-stage training | 0.722 | 0.719 | 0.680 | 0.788 | 0.784 |

Table VIII: **Joint vs. two-stage use of the proxy.** Joint training with the proxy as an auxiliary loss consistently yields the best performance.

**Training progression with and without the proxy.** We next focus on CLIP and study how the benefit of joint training evolves over training time. Table IX reports success rates after 5, 10, 15, 20, 25, and 30 epochs for direct fine-tuning versus joint training. Direct fine-tuning improves initially but tends to saturate (and even slightly fluctuate) beyond 20 epochs. In contrast, joint training not only starts from a higher performance but also continues to provide robust gains as training proceeds, yielding a consistent margin over direct fine-tuning at all checkpoints.

| Method | 5 ep. | 10 ep. | 15 ep. | 20 ep. | 25 ep. | 30 ep. |
|---|---|---|---|---|---|---|
| Direct fine-tuning | 0.573 | 0.680 | 0.732 | 0.765 | 0.771 | 0.769 |
| Joint training | **0.613** | **0.708** | **0.774** | **0.801** | **0.809** | **0.811** |

Table IX: **Training progression of CLIP with and without joint learning.** Success rates of CLIP under direct fine-tuning versus joint training across different epochs. Joint training consistently yields higher performance and continues to improve with longer training.

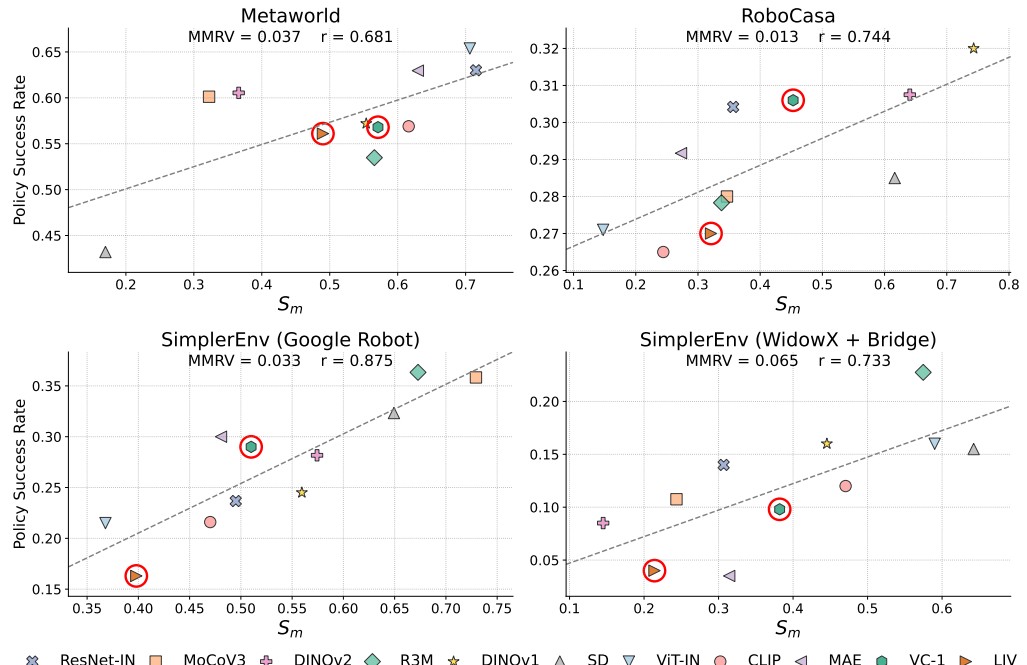

Figure II: **Correlation plots with robotics-specific encoders.** Our proxy metric remains predictive of the downstream policy success rate.

## A.8 ADDITIONAL ROBOTICS-SPECIFIC ENCODERS

We evaluate two additional robotics-specific encoders, VC-1 Majumdar et al. (2023b) and LIV Ma et al. (2023), under the same protocol. Both models are pretrained on large-scale robotics data and are designed to capture control-relevant structure directly from images. Figure II summarizes their policy success rates and correlation with proxy scores. VC-1 and LIV fall on the same correlation trend as our main models: higher state-prediction quality reliably corresponds to higher policy success, further supporting the generality of our proxy beyond standard image backbones.

## A.9 RELIANCE ON SIMULATION ENVIRONMENT STATES

Our proxy computation in the main paper uses simulator ground-truth states. We study how sensitive the method is to this choice and whether it can still work with noisy, estimated real-world states. We compare four variants of the proxy signal: (i) simulation proxy scores computed from ground-truth simulator states, (ii) real-world estimated proxy scores computed from estimated states (DetAny3D detections, robot states, and known object attributes), (iii) an action MSE proxy, and (iv) a depth-based proxy. For each variant, we measure its agreement with the true policy ranking using MMRV and Pearson's correlation (defined in the main paper). All metrics are averaged over 5 runs.

Tab. X reports the results. Using estimated real-world states leads to a slightly higher MMRV and somewhat lower Pearson correlation than simulator states, but remains substantially better than the action-MSE and depth-based baselines. Despite being noisy and incomplete, the estimated states

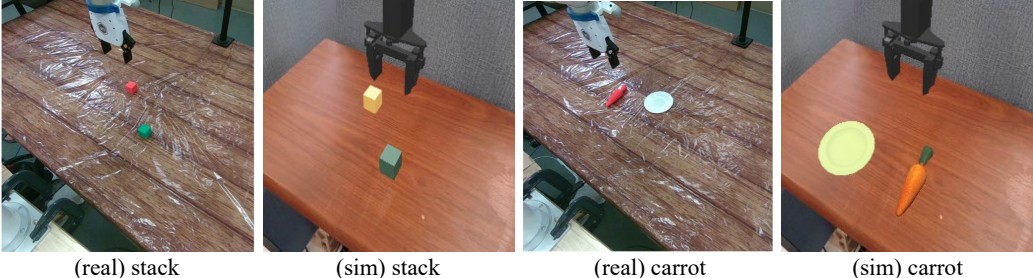

| (real) stack | (sim) stack | (real) carrot | (sim) carrot |

Figure III: **Domain gap between simulation and real-world environments. While the tasks are the same the environments differ significantly in appearance, object shapes, robot embodiment and camera poses.**

still yield a proxy that correlates well with policy performance. This suggests that the effectiveness of our proxy does not critically rely on privileged simulator information, and that similar signals could be obtained from real-world perception pipelines. In addition, though The simulation and real environments differ substantially in appearance, lighting, and layout (Figure III), the correlation remains high.

|  | MMRV ↓ | Pearson's correlation ↑ |
|---|---|---|
| Simulation proxy scores | 0.041 | 0.711 |
| Real-world estimated proxy scores | 0.047 | 0.532 |
| Action MSE | 0.055 | 0.347 |
| Depth | 0.080 | -0.372 |

Table X: **Effect of real-world proxies.** Estimated real-world states yield a proxy that is close to the simulator-based one and clearly better than action MSE or depth.

## A.10 DINOv1 AND DINOv2

DINOv1 (Caron et al., 2021) and DINOv2 (Oquab et al., 2024) learn different types of visual representations due to their training methodologies and data. DINOv1's self-supervised features tend to be more object-centric and segmentation-like, which can directly benefit tasks where identifying distinct objects is crucial (e.g. RoboCasa). DINOv2, while more powerful overall, captures a wider range of visual information, including emphasizing fine-grained spatial details, which may require fine-tuning to fully adapt to a specific task. As our experiments in section A.1 demonstrate, DINOv2 benefits more from fine-tuning compared to DINOv1, because its richer, more general features can then be specialized to the task.

## A.11 FAILURE CASES

While our proxy generally aligns well with policy performance, we have observed occasional outlier cases where it fails to predict the correct ranking. A notable example is MAE in the WidowX + Bridge setting (in Figure 4), where the aggregated proxy score overestimates its true performance.

This discrepancy arises when a model performs well on a small subset of state variables that are weakly correlated with task success, leading to an inflated average proxy score. For instance, MAE ranks highest in predicting pose and shape states—both identified in Table 2 as having low correlation with policy success—but ranks poorly across more informative state variables. As a result, its high proxy score does not reflect actual performance.

While such failure modes exist, we emphasize that they are rare. Across the benchmarks we study, the proxy consistently provides a reliable and interpretable signal: models with strong proxy scores typically demonstrate strong regression ability across relevant states and higher policy success. We will clarify these edge cases and their implications in the final version of the manuscript.

## B DETAILS OF BASELINE PROXIES

In this section, we detail the settings of other commonly used proxy objectives and analyze their latency bottlenecks.

**ImageNet Recognition.** Linear probing on ImageNet (Deng et al., 2009) is a well-established benchmark for representation quality. We freeze backbones and train a $k$-NN classifier (Hu et al., 2023) on the ImageNet 2012 train split, using the top-1 accuracy on the validation set as a proxy.

**Shape Bias.** Measuring shape bias on Stylized-ImageNet (Geirhos et al., 2018) evaluates a model's reliance on structural cues rather than superficial textures. We compute the fraction of shape-based predictions on the Stylized-ImageNet validation set using the same $k$-NN classification.

**Few-Shot Learning.** Few-shot learning is a popular approach to test the generalizability of a model. Here, we use only five demonstrations per task for training efficiency and measure the final performance. The success rate at test time is used as the proxy score. Aiming for efficiency, we only use five episodes when computing the success rate for each task.

**Action MSE.** We use the action prediction mean squared error loss (MSE) of the behavior cloning algorithm on the validation set as a proxy of its success rate. This approach requires training the behavior cloning head till convergence, but forgoes expensive policy execution. The training setting is the same as the Few-shot Learning above.

**Segmentation.** Semantic segmentation performance tests spatial and contextual feature learning (Burns et al., 2024; Man et al., 2024). To build the data for each task, we generate the trajectory together with the object mask in the simulator. The dataset is divided into a training and validation set for evaluation. We attach a lightweight segmentation head to the frozen encoder and report the mean Intersection-over-Union (mIoU) on the validation split.

**Depth Estimation.** Monocular depth estimation probes geometric reasoning (Banani et al., 2024). Similarly to segmentation, we render the depth for each trajectory as the objective. A small depth decoder is attached to each frozen backbone, and we evaluate the MSE on the validation set.

## C DETAILS OF COMPLEXITY AND LATENCY COMPARISON

### C.1 ANALYSIS OF RESNET-IN AND VIT-IN BACKBONES

The variation in latency delta between ResNet-IN and ViT-IN across proxy tasks in Table 4 arises not only from architectural differences (e.g., ResNet vs. ViT) but also from the nature of each proxy task's training and evaluation pipeline. While the backbones are frozen, the computation time depends on several additional factors such as head architecture, batch size, data scale, and whether inference or rollout is involved. Below is a breakdown of the main factors affecting latency across task categories.

**Few-Shot Policy Learning** : This setting involves full policy training from a few demonstrations using a Diffusion Policy that includes a frozen visual encoder, a text encoder, and a 1D diffusion network. The total time includes both training and environment rollouts. The rollout and diffusion steps contribute significantly to the overall time, and differences between ResNet and ViT (e.g., in feature dimensionality and processing overhead) become more pronounced in this larger pipeline.

**Action MSE** : This setup shares the same architecture as Few-Shot (including the full diffusion model), but evaluation is done purely on the validation set without rollouts. This leads to a smaller runtime overall and a reduced latency gap between backbones, since the rollout overhead is removed and the backbone usage is more uniform.

**ImageNet** : Feature extraction and evaluation over the full ImageNet validation set.

**Shape Bias** : Requires two passes over the dataset (to isolate texture and shape cues), which effectively doubles the total runtime, and hence the latency delta between ResNet and ViT also roughly doubles.

**Depth & Segmentation** : These are training-based dense prediction tasks. A lightweight decoder head is trained on top of the frozen encoder, and the additional runtime is dominated by the process of high-resolution feature maps being upsampled to the input resolution, and MLP predictors being applied to the large feature maps.

In summary, the inconsistencies in the latency delta reflect the interaction between the frozen backbone and the downstream task-specific computation, including head architecture, data scale, and whether inference includes rollouts or dense pixel-wise outputs.

## C.2 TIME BREAKDOWN ANALYSIS

In addition, to have a better assessment of the time, we give the breakdown of time consumption here:

**Few-Shot Learning**

ResNet: Policy training (38 min), rollout evaluation (40 min)

ViT: Policy training (61 min), rollout evaluation (52 min)

**Action MSE**

ResNet: Policy training (38 min), evaluation (7 min)

ViT: Policy training (61 min), evaluation (11 min)

**ImageNet**

ResNet: Feature extraction (9 min), KNN prediction (2 min)

ViT: Feature extraction (41 min), KNN prediction (5 min)

**Shape Bias**

ResNet: Feature extraction (20 min), KNN prediction (3 min)

ViT: Feature extraction (81 min), KNN prediction (11 min)

**Segmentation & Depth Estimation**

ResNet: Training (10 min), evaluation (0.2 min)

ViT: Training (27 min), evaluation (0.3 min)

**State Regression (Ours)**

ResNet: Training (4 min), evaluation (0.2 min)

ViT: Training (12 min), evaluation (0.3 min)

## D SIMULATION ENVIRONMENTS

### D.1 ENVIRONMENT DETAILS AND EXPERT POLICY

**Simulation environments.** We evaluate four distinct manipulation benchmarks, each with different task counts, object diversity, difficulty levels, and robot kinematics:

1. **MetaWorld (50 tasks)** (Yu et al., 2020)
   - *Task count:* 50 distinct tabletop manipulation tasks.
   - *Object diversity:* tasks involve one or two CAD-style objects drawn from a set of prototypical meshes (doors, drawers, windows, buttons, pegs, boxes, etc.).
   - *Difficulty:* spans *easy* (reach, push), *medium* (pick–and–place, press button), to *hard* (peg-in-hole, open window). Since all tasks use the same scene setting, this benchmark is relatively *easy*.
   - *Robot:* Rethink Sawyer 7-DoF arm with parallel-jaw gripper.
   - *Action control:* continuous 4-dimensional end-effector displacement $(\Delta x, \Delta y, \Delta z)$ plus gripper open/close command.
2. **RoboCasa (24 tasks)** (Nasiriany et al., 2024)
   - *Task count:* 100 everyday household tasks in kitchen scenes. We adopt the 24 atomic tasks in the experiments.
   - *Object diversity:* over 2,500 3D assets spanning 150+ object categories (cabinets, appliances, utensils, etc.).

- *Difficulty:* includes *atomic* tasks (pick, place) and *composite* multi-step tasks (e.g., cooking sequences), ranging from *medium* to *high*.
- *Robot:* Rethink Sawyer 7-DoF arm (we instantiate Sawyer in all scenes).
- *Action control:* 7-dimensional controller commanding end-effector pose ($x, y, z$,yaw, pitch, roll) plus gripper open/close.

3. **SimplerEnv (Li et al., 2024) (G) (10 tasks)**
   - *Task count:* 10 pick tasks, including 6 tasks for the Google Robot and 4 tasks for the WidowX.
   - *Object diversity:* different household items (tools, blocks, bottles).
   - *Difficulty: medium*, as each goal object is randomly sampled.
   - *Robot:* Fetch Robotics arm (7-DoF articulated manipulator) for Google Robot and WidowX-250 S 6-DoF research arm for WidowX.
   - *Action control:* 7-dimensional controller commanding end-effector pose ($x, y, z$,yaw, pitch, roll) plus gripper open/close.

**Demonstration and state collection.** As mentioned in the paper, we collect demonstrations following a different strategy in each simulation environment. Specifically, for MetaWorld, we use their expert policy to generate the demonstrations. For RoboCasa, we use official human-collected demonstrations. In SimplerEnv, we design the expert policy by ourselves and generate the demonstrations. Specifically, the expert policy is built with observable goal states. For each task, we write its corresponding goal-conditioned manipulation code and validate our expert policy through the success rate. Approximately, our expert policy can reach about 60% success rates.

Regarding the state information, robot-level states like joints and end-effector states can be directly gathered from the simulator. In terms of lighting, we give different lighting conditions a unique class identifier. The object states are stored differently across the environments. In MetaWorld, we exclude the background objects and retrieve the foreground MuJoCo bodies. In RoboCasa, the foreground objects are directly stored in the environment information. In SimplerEnv, the foreground objects are considered dynamic "actors" from the environment. Once we have the object instance, we can extract its pose and material states, and the shape is defined using the united bounding box for the object's parts. The segmentation mask is rendered from the simulator. If the object has no segmentation mask rendered, it is considered not observable in the current view. The bounding box is generated from the segmentation mask.

# E IMPLEMENTATION DETAILS

## E.1 PRETRAINED VISION MODELS

**Overview.** We consider a diverse set of pretrained visual models widely adopted for downstream visual tasks, particularly in robotics and embodied AI. Each model differs in backbone architecture, training objective, and pretraining dataset. Table XI provides an overview.

**Feature extraction details.** Our feature extraction for visual backbones is split into three categories: CNN-based, ViT-based, and diffusion-based. The specifics are as follows:

1. **CNN-based (ResNet-18).**
   - *Feature map:* output of the final convolutional block (pre-pooling), size $B \times C \times H \times W$.
   - *Global feature:* average-pooled vector of size $B \times C$.

2. **ViT-based (ViT-B).**
   - *Feature map:* reshape the remaining $N$ patch embeddings into $B \times C \times \sqrt{N} \times \sqrt{N}$.
   - *Global feature:* the learned class-token embedding, shape $B \times C$.

3. **Diffusion-based (Stable Diffusion v1.5).**
   - *Feature map:* extract intermediate feature maps from selected up-sampling blocks, each of shape $B \times C_i \times h_i \times w_i$.
   - *Global feature:* spatially average one map into a $B \times C_i$ descriptor.

| Name | Backbone Setting (Param) | Loss Function | Pretrained Data (Size) |
|---|---|---|---|
| ResNet-IN | ResNet-18 12M | Supervised cross-entropy | ImageNet-1k (∼1.3M images) |
| ViT-IN | ViT-B/16 (86M) | Supervised cross-entropy | ImageNet-1k (∼1.3M images) |
| CLIP | ViT-B (86M) | Contrastive image-text alignment | OpenAI curated (∼400M images) |
| DINOv1 | ViT-B/16 (86M) | Self-distillation w/o labels | ImageNet-1k (∼1.3M images) |
| DINOv2 | ViT-B/14L (86M) | Self-distillation with momentum teacher | Internal curated set (∼142M images) |
| MoCo v3 | ViT-B/16 (86M) | Contrastive learning (MoCo) | ImageNet-1k (∼1.3M images) |
| MAE | ViT-B/16 (86M) | Masked image reconstruction | ImageNet-1k (∼1.3M images) |
| R3M | ResNet-18 (12M) | Contrastive w/ robot goal-conditioning | Ego4D, HM3D, etc. (∼100M+ frames) |
| SD | U-Net w/ ViT encoder (∼860M) | Denoising diffusion objective | LAION-5B (∼5B images) |

Table XI: Summary of pretrained visual models. Param: number of parameters. Dataset size is approximate.

## E.2 LICENSE OF DATASETS AND SIMULATOR USED

We list the licenses of all the simulation environments we have used during our experiments.

- Meta-World (Yu et al., 2020): MIT License.
- RoboCasa (Nasiriany et al., 2024): MIT License.
- SimplerEnv (G & W) (Li et al., 2024): MIT License.

In addition, we use the official implementation and pre-trained models provided on the Huggingface, GitHub, and Pytorch Hub for ResNet-IN[*], ViT-IN[†], CLIP[‡], DINOv1[§], DINOv2[¶], MoCo v3[‖], MAE v3[**], R3M[††] and SD[‡‡]. All of the foundation models we evaluated in our paper utilize various data sources during their pre-training phase. Please refer to their original paper for the license of the datasets they have used in pre-training their models.

## E.3 ADDITIONAL DETAILS OF THE STATE REGRESSION PROXY

To make a fair and straightforward comparison between different backbones, we unify the state regression with the same resolution of the feature map. Specifically, the feature maps are interpolated to $7 \times 7$ resolution and later used for the RoI-pooling. For the optimization hyperparameter, we train the model for 20 epochs with 32 batch size. We use the SGD optimizer and set the learning rate to 5e-4. The momentum is set to 0.9, and the weight decay is set to 1e-4.

---

[*] https://docs.pytorch.org/vision/main/models/resnet.html
[†] https://docs.pytorch.org/vision/main/models/vision_transformer.html
[‡] https://github.com/mlfoundations/open_clip
[§] https://huggingface.co/timm/vit_small_patch16_224.dino
[¶] https://github.com/facebookresearch/dinov2
[‖] https://github.com/facebookresearch/moco-v3
[**] https://huggingface.co/timm/vit_base_patch16_224.mae
[††] https://github.com/facebookresearch/r3m
[‡‡] https://huggingface.co/stable-diffusion-v1-5/stable-diffusion-v1-5

## F    LIMITATION AND BROADER IMPACT

**Limitation.**    A central limitation of our approach is its reliance on access to ground-truth environment state, which is readily available in simulation but not necessarily in the real world. While simulators are a widely accepted tool for efficient and reproducible evaluation in robotics, they can introduce a domain gap, and some natural phenomena, like splashing water, or smoke are very challenging to accurately capture with existing simulators. Furthermore, our analysis is limited to three simulation environments and a finite set of pretrained visual backbones; while diverse, this does not capture the full variability of robotic tasks or vision models. Finally, although our method is computationally efficient relative to baselines, it still requires learning state regression heads.

**Broader Impact.**    Our method is not directly tied to any specific deployment, and we do not anticipate immediate societal risks from the method itself. However, improved tools for evaluating and scaling robotic policies may accelerate the deployment of autonomous systems, which in turn can have significant societal implications (both positive and negative). Our reliance on simulation also raises concerns about potential mismatches between training conditions and real-world scenarios, which could lead to unexpected behavior if not properly validated. We encourage researchers to complement our approach with robustness checks in real-world environments.

## G    USE OF LARGE LANGUAGE MODELS (LLMS)

We used Large Language Models (LLMs) only to facilitate writing, such as polishing grammar, improving clarity, and suggesting alternative phrasings for text drafted by the authors. LLMs were not used for research ideation or to generate core scientific content. All LLM-assisted text was reviewed and edited by the authors before inclusion.

## H    SUCCESS RATE BREAKDOWN

In this section, we breakdown the success rate for different backbones in Figure 3. The results are shown in Table XII, XIII, XIV and XV.

| Task | ResNet-IN | R3M | ViT-IN | MoCov3 | DINOv1 | DINOv2 | MAE | CLIP | SD |
|---|---|---|---|---|---|---|---|---|---|
| assembly | 0.65 | 0.19 | 0.51 | 0.44 | 0.47 | 0.33 | 0.47 | 0.37 | 0.11 |
| basketball | 0.18 | 0.12 | 0.15 | 0.10 | 0.09 | 0.06 | 0.14 | 0.07 | 0.05 |
| bin-picking | 0.45 | 0.18 | 0.66 | 0.35 | 0.81 | 0.56 | 0.29 | 0.86 | 0.08 |
| box-close | 0.57 | 0.58 | 0.64 | 0.53 | 0.60 | 0.50 | 0.69 | 0.53 | 0.33 |
| button-press-topdown | 0.96 | 1.00 | 1.00 | 0.91 | 0.94 | 0.86 | 0.99 | 0.88 | 0.79 |
| button-press-topdown-wall | 0.95 | 1.00 | 1.00 | 0.95 | 0.92 | 0.88 | 0.99 | 0.73 | 0.66 |
| button-press | 1.00 | 0.92 | 1.00 | 0.94 | 0.92 | 0.87 | 1.00 | 0.84 | 0.86 |
| button-press-wall | 1.00 | 1.00 | 1.00 | 1.00 | 1.00 | 0.99 | 1.00 | 0.98 | 0.91 |
| coffee-button | 0.95 | 0.95 | 0.98 | 1.00 | 0.86 | 0.97 | 1.00 | 0.72 | 0.86 |
| coffee-pull | 0.44 | 0.00 | 0.46 | 0.37 | 0.27 | 0.33 | 0.48 | 0.39 | 0.13 |
| coffee-push | 0.30 | 0.15 | 0.42 | 0.45 | 0.27 | 0.31 | 0.53 | 0.25 | 0.11 |
| dial-turn | 0.26 | 0.02 | 0.37 | 0.36 | 0.16 | 0.45 | 0.25 | 0.21 | 0.01 |
| disassemble | 0.43 | 0.23 | 0.50 | 0.39 | 0.40 | 0.09 | 0.45 | 0.38 | 0.12 |
| door-close | 1.00 | 1.00 | 1.00 | 1.00 | 1.00 | 1.00 | 1.00 | 1.00 | 1.00 |
| door-lock | 0.93 | 0.87 | 0.41 | 0.67 | 0.64 | 0.69 | 0.78 | 0.59 | 0.55 |
| door-open | 1.00 | 1.00 | 1.00 | 1.00 | 0.99 | 1.00 | 1.00 | 1.00 | 0.88 |
| door-unlock | 1.00 | 0.88 | 0.95 | 0.91 | 0.81 | 0.89 | 0.97 | 0.85 | 0.13 |
| hand-insert | 0.18 | 0.14 | 0.31 | 0.28 | 0.35 | 0.37 | 0.33 | 0.31 | 0.16 |
| drawer-close | 1.00 | 1.00 | 1.00 | 1.00 | 1.00 | 1.00 | 1.00 | 1.00 | 1.00 |
| drawer-open | 1.00 | 0.99 | 0.86 | 0.95 | 0.90 | 0.98 | 0.83 | 0.93 | 0.59 |
| faucet-open | 1.00 | 1.00 | 1.00 | 1.00 | 1.00 | 1.00 | 1.00 | 1.00 | 1.00 |
| faucet-close | 1.00 | 1.00 | 1.00 | 0.92 | 0.99 | 0.95 | 1.00 | 0.94 | 0.92 |
| hammer | 0.37 | 0.32 | 0.50 | 0.38 | 0.27 | 0.43 | 0.50 | 0.45 | 0.26 |
| handle-press-side | 1.00 | 1.00 | 1.00 | 1.00 | 1.00 | 1.00 | 1.00 | 1.00 | 0.97 |
| handle-press | 0.99 | 1.00 | 1.00 | 1.00 | 0.98 | 0.98 | 1.00 | 1.00 | 0.67 |
| handle-pull-side | 0.51 | 0.27 | 0.52 | 0.58 | 0.55 | 0.45 | 0.64 | 0.61 | 0.07 |
| handle-pull | 0.03 | 0.00 | 0.11 | 0.07 | 0.22 | 0.11 | 0.05 | 0.01 | 0.01 |
| lever-pull | 0.54 | 0.09 | 0.39 | 0.61 | 0.32 | 0.60 | 0.60 | 0.25 | 0.08 |
| pick-place-wall | 0.28 | 0.17 | 0.38 | 0.49 | 0.20 | 0.52 | 0.29 | 0.25 | 0.15 |
| pick-out-of-hole | 0.37 | 0.02 | 0.63 | 0.17 | 0.16 | 0.32 | 0.58 | 0.22 | 0.32 |
| pick-place | 0.04 | 0.01 | 0.18 | 0.21 | 0.04 | 0.14 | 0.04 | 0.02 | 0.01 |
| plate-slide | 0.86 | 1.00 | 0.79 | 0.73 | 0.76 | 0.81 | 0.87 | 0.70 | 0.63 |
| plate-slide-side | 1.00 | 1.00 | 1.00 | 1.00 | 1.00 | 1.00 | 1.00 | 1.00 | 1.00 |
| plate-slide-back | 0.99 | 1.00 | 1.00 | 1.00 | 0.95 | 0.97 | 1.00 | 1.00 | 0.99 |
| plate-slide-back-side | 1.00 | 1.00 | 1.00 | 1.00 | 1.00 | 1.00 | 1.00 | 1.00 | 1.00 |
| peg-insert-side | 0.24 | 0.28 | 0.25 | 0.21 | 0.19 | 0.19 | 0.40 | 0.18 | 0.07 |
| peg-unplug-side | 0.78 | 0.48 | 0.68 | 0.71 | 0.53 | 0.59 | 0.62 | 0.47 | 0.16 |
| soccer | 0.15 | 0.07 | 0.14 | 0.09 | 0.15 | 0.10 | 0.16 | 0.07 | 0.03 |
| stick-push | 0.71 | 0.98 | 0.96 | 0.38 | 0.52 | 0.82 | 0.74 | 0.80 | 0.59 |
| stick-pull | 0.33 | 0.33 | 0.30 | 0.43 | 0.28 | 0.44 | 0.28 | 0.53 | 0.30 |
| push | 0.35 | 0.02 | 0.36 | 0.19 | 0.08 | 0.34 | 0.28 | 0.18 | 0.10 |
| push-wall | 0.47 | 0.07 | 0.72 | 0.70 | 0.45 | 0.60 | 0.38 | 0.36 | 0.28 |
| push-back | 0.31 | 0.05 | 0.38 | 0.19 | 0.22 | 0.31 | 0.25 | 0.27 | 0.11 |
| reach | 0.25 | 0.34 | 0.24 | 0.16 | 0.19 | 0.22 | 0.20 | 0.16 | 0.10 |
| reach-wall | 0.71 | 0.44 | 0.57 | 0.63 | 0.52 | 0.65 | 0.52 | 0.50 | 0.37 |
| shelf-place | 0.30 | 0.18 | 0.15 | 0.12 | 0.11 | 0.08 | 0.09 | 0.07 | 0.03 |
| sweep-into | 0.35 | 0.21 | 0.50 | 0.28 | 0.32 | 0.32 | 0.37 | 0.35 | 0.19 |
| sweep | 0.32 | 0.19 | 0.73 | 0.21 | 0.25 | 0.32 | 0.48 | 0.22 | 0.07 |
| window-open | 1.00 | 1.00 | 0.99 | 1.00 | 0.95 | 0.88 | 0.95 | 0.95 | 0.88 |
| window-close | 1.00 | 1.00 | 1.00 | 1.00 | 1.00 | 1.00 | 1.00 | 1.00 | 0.93 |

Table XII: Success rate breakdown for MetaWorld.

| Task | ResNet-IN | R3M | ViT-IN | MoCov3 | DINOv1 | DINOv2 | MAE | CLIP | SD |
|---|---|---|---|---|---|---|---|---|---|
| PnPCounterToCab | 0.00 | 0.02 | 0.00 | 0.00 | 0.00 | 0.02 | 0.00 | 0.00 | 0.00 |
| PnPCabToCounter | 0.04 | 0.04 | 0.04 | 0.00 | 0.04 | 0.02 | 0.00 | 0.02 | 0.00 |
| PnPCounterToSink | 0.02 | 0.02 | 0.00 | 0.00 | 0.02 | 0.02 | 0.00 | 0.02 | 0.00 |
| PnPSinkToCounter | 0.00 | 0.00 | 0.00 | 0.02 | 0.04 | 0.02 | 0.02 | 0.02 | 0.04 |
| PnPCounterToMicrowave | 0.00 | 0.00 | 0.00 | 0.02 | 0.06 | 0.02 | 0.04 | 0.08 | 0.02 |
| PnPMicrowaveToCounter | 0.00 | 0.00 | 0.00 | 0.06 | 0.02 | 0.04 | 0.04 | 0.00 | 0.00 |
| PnPCounterToStove | 0.00 | 0.00 | 0.02 | 0.02 | 0.00 | 0.02 | 0.00 | 0.02 | 0.02 |
| PnPStoveToCounter | 0.00 | 0.00 | 0.00 | 0.00 | 0.00 | 0.00 | 0.00 | 0.02 | 0.02 |
| OpenSingleDoor | 0.32 | 0.32 | 0.28 | 0.26 | 0.48 | 0.28 | 0.26 | 0.16 | 0.16 |
| CloseSingleDoor | 0.56 | 0.58 | 0.62 | 0.70 | 0.62 | 0.68 | 0.80 | 0.70 | 0.70 |
| OpenDoubleDoor | 0.16 | 0.12 | 0.30 | 0.18 | 0.30 | 0.12 | 0.30 | 0.12 | 0.32 |
| CloseDoubleDoor | 0.62 | 0.60 | 0.48 | 0.54 | 0.64 | 0.62 | 0.74 | 0.54 | 0.52 |
| OpenDrawer | 0.58 | 0.56 | 0.58 | 0.44 | 0.46 | 0.58 | 0.46 | 0.40 | 0.56 |
| CloseDrawer | 0.98 | 0.92 | 0.96 | 0.92 | 0.88 | 0.90 | 0.96 | 0.96 | 0.98 |
| TurnOnSinkFaucet | 0.44 | 0.50 | 0.46 | 0.46 | 0.54 | 0.54 | 0.42 | 0.46 | 0.36 |
| TurnOffSinkFaucet | 0.64 | 0.72 | 0.56 | 0.76 | 0.84 | 0.70 | 0.58 | 0.60 | 0.68 |
| TurnSinkSpout | 0.62 | 0.46 | 0.50 | 0.58 | 0.62 | 0.60 | 0.62 | 0.64 | 0.62 |
| TurnOnStove | 0.58 | 0.42 | 0.28 | 0.38 | 0.44 | 0.48 | 0.50 | 0.56 | 0.40 |
| TurnOffStove | 0.10 | 0.14 | 0.08 | 0.08 | 0.10 | 0.10 | 0.08 | 0.06 | 0.08 |
| CoffeeSetupMug | 0.02 | 0.02 | 0.06 | 0.08 | 0.04 | 0.08 | 0.02 | 0.00 | 0.02 |
| CoffeeServeMug | 0.32 | 0.06 | 0.16 | 0.30 | 0.24 | 0.28 | 0.10 | 0.10 | 0.08 |
| CoffeePressButton | 0.24 | 0.24 | 0.28 | 0.26 | 0.24 | 0.14 | 0.16 | 0.34 | 0.22 |
| TurnOnMicrowave | 0.54 | 0.42 | 0.42 | 0.30 | 0.50 | 0.40 | 0.26 | 0.28 | 0.60 |
| TurnOffMicrowave | 0.52 | 0.54 | 0.40 | 0.36 | 0.56 | 0.72 | 0.64 | 0.26 | 0.46 |

Table XIII: Success rate breakdown for RoboCasa.

| Task | ResNet-IN | R3M | ViT-IN | MoCov3 | DINOv1 | DINOv2 | MAE | CLIP | SD |
|---|---|---|---|---|---|---|---|---|---|
| pick_coke_can | 0.35 | 0.53 | 0.33 | 0.58 | 0.37 | 0.31 | 0.34 | 0.15 | 0.16 |
| pick_object | 0.11 | 0.21 | 0.14 | 0.16 | 0.09 | 0.12 | 0.15 | 0.07 | 0.08 |
| move_near | 0.05 | 0.07 | 0.07 | 0.10 | 0.06 | 0.05 | 0.09 | 0.10 | 0.04 |
| open_drawer | 0.53 | 0.64 | 0.40 | 0.62 | 0.54 | 0.57 | 0.63 | 0.50 | 0.77 |
| close_drawer | 0.37 | 0.68 | 0.34 | 0.62 | 0.39 | 0.63 | 0.58 | 0.48 | 0.88 |
| place_in_closed_top_drawer | 0.01 | 0.05 | 0.01 | 0.07 | 0.02 | 0.01 | 0.01 | 0.00 | 0.01 |

Table XIV: Success rate breakdown for SimplerEnv (Google Robot).

| Task | ResNet-IN | R3M | ViT-IN | MoCov3 | DINOv1 | DINOv2 | MAE | CLIP | SD |
|---|---|---|---|---|---|---|---|---|---|
| spoon_on_towel | 0.12 | 0.11 | 0.10 | 0.10 | 0.05 | 0.00 | 0.00 | 0.03 | 0.12 |
| carrot_on_plate | 0.01 | 0.09 | 0.01 | 0.03 | 0.02 | 0.08 | 0.01 | 0.01 | 0.10 |
| stack_cube | 0.05 | 0.11 | 0.03 | 0.02 | 0.04 | 0.00 | 0.03 | 0.08 | 0.02 |
| put_eggplant_in_basket | 0.38 | 0.60 | 0.50 | 0.28 | 0.53 | 0.26 | 0.1 | 0.36 | 0.38 |

Table XV: Success rate breakdown for SimplerEnv (WidowX + Bridge).

