# OpenReview forum: "Capturing Visual Environment Structure Correlates with Control Performance"
_ICLR.cc/2026/Conference — ICLR 2026 Poster_

### Official Review · Reviewer_oHEz · 2025-10-16

**Soundness:** 3
**Presentation:** 3
**Contribution:** 3
**Rating:** 4
**Confidence:** 4

**Summary:**

The authors propose a new proxy for manipulation downstream policy success by measuring how well visual encoder pretrained backbones can predict the state of the environment (lighting conditions, robot arm joints and end effector pose, objects' positions, orientations, shapes, materials). Training such regressors is possible in simulation where ground truth state is available. For a plethora of backbone encoders, the proxy is shown to provide significantly better correlation to task performance than other methods both in simulation and on real world deployment. In addition, jointly fine-tuning visual backbones (5 different variants) for policy learning as well as state prediction is shown to  consistently offer clear improvements to policy performance on the MetaWorld benchmark.

**Strengths:**

The paper is well presented and thorough in situating the work within the current literature.
The manuscript is clear and enjoyable to read with a logical and consistent progression.

The solution presented to the proxy measure learning problem is simple and intuitive which makes it convincing. Its implementation in terms of encoding, multi-object handling, loss design and metric definition are all clear and logical.

The experimental protocol is thorough, at least for the base experiments looking into correlation quality, with many relevant benchmarks and visual encoder backbones evaluated both in simulation and in the real world.

The results appear to back the claims of the paper very elegantly.

**Weaknesses:**

**On the evaluation protocol:**
- The authors state at line 302 that the protocol follows that of the SimplerEnv benchmark without any additional details or what part of the evaluation this entails, one suspects it is the averaged success rate measurements but that is not clear
- Furthermore, the authors select the Mean Maximum Rank Violation (MMRV) metric as well as the Pearson correlation between performance and state prediction capacity as the statistical measures of correlation without any introduction, explanation or presentation of the what these metrics represent and why those and not others are utilized in the work. This leaves room for doubt as to whether the metric selection is cherry picked to amplify the paper results or objectively standard. This warrants more clarity as the choice is not explained in the main text, nor is it clarified in the appendix.

**On the results:**
- The success rates vary at most by about 15% percent between the best and worst models, with 3 out of four benchmarks having all models perform at low success rates. Each datapoint is obtained with 100 rollouts. Looking only at the MMRV and r numbers provided (and not explained as mentioned above) and average policy success rates with no standard deviations with different x axis scales, it is difficult to appreciate the statistical significance of the results.

**On the analysis of individual state dimensions:**

I thought this was a very nice ablation to have, yet it is not easy to decipher how it was conducted and what is interesting about the results. After some scrolling back and forth my understanding is that from the full state regressor you extract the specific predicted state entries and compute the MMRV score wrt performance. The idea being that the ones with the lowest would correlate the best with performance should they have been used on their own. In many ways this feels incomplete and perhaps the wrong question to ask:
- The state dimensions are hand designed, and although they seem reasonable one could argue that the possibilities/variants might extend to other attributes. If the authors assume that training a regressor on the full state is the best approach, they should make a case that shows that each attribute or dimension in the state is actually contributing to correlation (at least for some backbone/benchmark combination)
- Otherwise the authors should be looking for the best mix of attributes to consider for correlation and considering the couplings that arise both in the training of regressors as well as in establishing the correlation scores.
- Indeed, with numbers varying quite aggressively across benchmarks for a single state dimension on its own, the only conclusion is that (l 458) "that different environments indeed present different demands for visual representations". But what about doing a leaving one out experiment for example to try to understand not what each brings on its own, but which are not that useful in the mix.
- In summary, it is unclear whether all the attributes in the full state regressor are needed and how this affects the state regression quality should they be reduced/modified which can then couple back into the correlation quality.

**Application demonstration asymmetry :**
- First and foremost, though well written, the paper fails to my taste to clearly explain what are the uses of such proxies and why they are important tools. They might be cheaper to evaluate during architecture design or backbone selection for a policy but the case could be. made clearer as to their exact purpose.
- The most interesting application beyond ranking seems to be joint fine-tuning which the authors show provides substantial performance gains across backbone models. This is a very interesting aspect of the work that is largely neglected compared to the potential interest for practitioners. This part of the work warrants more results and details.

The authors have omitted to include the use of LLMs statement.

**Questions:**

- Are all the tasks considered on parallel gripper pick and place tasks?

- This circles back to points about what state attributes to consider, for tasks with end effector camera where the arm body is not visible how would you say things would differ, one can only predict general scene and object attributes in this situation. Would similar correlations hold in your opinion?

- The other proxies seem very generic, are there no intermediate metrics or any works that provide proxies that are closer to capturing the state of the scene?

- Along these lines, are there ways to combine the pre-existing metrics to see if some combination of them can challenge your approach?

---

> ### Author Response · Authors · 2025-11-24
> **Response to Reviewer oHEz (Part 1)**
>
> Thank you for the thoughtful and constructive feedback. We are encouraged by your positive assessment of our method's effectiveness and intuitiveness and of the thoroughness of our empirical analysis. We respond to your questions and concerns individually below.
>
> **(1) Details of the experimental protocol.**
> Thank you for pointing this out. We acknowledge that the description of our evaluation protocol and correlation metrics lacked sufficient detail. In the revised version, we have clarified the components inherited from the SimplerEnv benchmark (including success rate computation) and added explanations of both MMRV and Pearson correlation in Section 4 (Evaluation Protocol). MMRV and Pearson Correlation metrics are adopted from prior work [A,B,C] and are standard for this type of correlation analysis in representation evaluation, rather than being specific to our study.
>
> - [A] Li, Xuanlin, et al. “Evaluating real-world robot manipulation policies in simulation.” In CoRL, 2024.
> - [B] Kadian, Abhishek, et al. “Sim2real predictivity: Does evaluation in simulation predict real-world performance?” IEEE Robotics and Automation Letters 5.4 (2020).
> - [C] Karl Pearson. “Note on regression and inheritance in the case of two parents.” Proceedings of the Royal Society of London, 58:240–242, 1895.
>
> **(2) Statistical significance of the results.**
> We have reported error bars for policy performance in the MetaWorld environment in Section A.4 in the Appendix. These results demonstrate that the Standard Error of the Mean is an **order of magnitude smaller** than the performance gaps between models, supporting the statistical robustness of our findings.
>
> Please note that each success rate represents an average across tasks, with 100 rollouts per task. For example, MetaWorld includes 50 tasks, resulting in 5,000 rollouts per encoder — a substantial sample size that keeps estimation noise negligible.
>
> **(3) Analysis of individual state dimensions.**
> We thank the reviewer for this detailed and constructive feedback. To clarify, the state dimensions are **not hand-designed**. Instead, our method uses *all* variables directly exposed by each simulator (e.g., object pose, joint angles, material properties) to define a uniform state representation, without introducing subjective choices about which factors “should” matter for each task. Accordingly, the per-dimension ablation is not intended to show that every individual attribute contributes to correlation, but to illustrate how different environments emphasize distinct parts of the state. We have clarified both the intent and scope of this analysis in Section 5.
>
> To address the reviewer’s suggestion, we performed two complementary analyses. Firstly, the leave-one-out experiment below shows that removing key states (e.g., joint angles, object pose) noticeably decreases correlation — for example, in MetaWorld r = 0.69 → 0.62 (w/o $q^J$), RoboCasa $r$ = 0.76 → 0.70 (w/o $s_{shape}$) — confirming that these dimensions are critical.
>
> | Environment / Metric   | w/o $p_{pose}$ | w/o $q_{pose}$ | w/o $s_{shape}$ | w/o $m_{mat}$ | w/o $q^J$ | w/o $p^{ee}$ | w/o $l$  | Full states |
> |---------|:-------------:|:-------------:|:--------------:|:------------:|:----------:|:-----------:|:---------:|:------------:|
> | MetaWorld ($r$)          | 0.7325       | 0.6358       | 0.7607        | 0.6979      | 0.6153    | 0.6656     | 0.6656   | 0.691       |
> | MetaWorld (MMRV)       | 0.0402       | 0.0455       | 0.0361        | 0.0365      | 0.0365    | 0.0308     | 0.0365   | 0.037       |
> | RoboCasa ($r$)           | 0.7169       | 0.7342       | 0.7028        | 0.7797      | 0.7744    | 0.7375     | 0.7602   | 0.760       |
> | RoboCasa (MMRV)        | 0.0163       | 0.0163       | 0.0186        | 0.0105      | 0.0153    | 0.0134     | 0.0099   | 0.010       |
>
> Secondly, to explore the best possible combination, we took the brute-force oracle approach and enumerated all the combinations, selecting the best one for each environment based on the Pearson correlation score. The results below demonstrate that manually selecting state dimensions shows moderate, environment-specific gains. However, the optimal subset varies across benchmarks.
>
> | Setting                         | MMRV ↓ | Pearson’s correlation ↑ |
> |--|:--:|:--:|
> | MetaWorld (full states)         | 0.037  | 0.691                    |
> | MetaWorld ($q_{pose}$, $q^{J}$)     | 0.025  | 0.870                    |
> | RoboCasa (full states)          | 0.010  | 0.760                    |
> | RoboCasa ($p_{pose}$, $s_{shape}$)  | 0.011  | 0.854                    |
>
> These findings demonstrate that while certain dimensions contribute more strongly than others, our **full-state formulation maintains competitive correlation** (within ≈10–15% of the best oracle subsets) while greatly simplifying the setup. Thus, our universal, task-agnostic approach achieves stable performance across diverse environments without per-task tuning.

---

> ### Author Response · Authors · 2025-11-24
> **Response to Reviewer oHEz (Part 2)**
>
> **(4) Generalization to different scenarios.**
> We appreciate this thoughtful question. All of our main experiments indeed focus on parallel-gripper manipulation, as these tasks are standard in the literature. However, our formulation is agnostic to the specific manipulator or viewpoint: it simply regresses the full set of simulator-exposed state variables. To test this, we evaluated our approach on an ego-centric **navigation** task in RoboCasa, where the camera is moving with the agent.
>
> Using the same diffusion policy implementation, visual backbones, and state vector definition as in the manipulation experiments, we obtained a similarly strong correlation between proxy predictions and policy performance. This result demonstrates that our metric remains reliable even when the available visual information differs substantially — supporting the claim that it generalizes across task types and viewpoints without modification.
>
> | Setting                               | MMRV ↓ | Pearson’s correlation ↑ |
> |------|:--:|:----:|
> | RoboCasa Manipulation (Ours)          | 0.010  | 0.760  |
> | RoboCasa Navigation (Ours)            | 0.022  | 0.727   |
> | RoboCasa Navigation (Segmentation)    | 0.037  | 0.436                    |
> | RoboCasa Navigation (Depth)           | 0.043  | 0.172                    |
>
> **(5) Applications.**
> We appreciate this valuable perspective. While our primary focus is on analyzing the representational factors that enable generalizable visual control, we agree that clarifying the practical role of our proxy is important. In the revised version, we have expanded the Introduction (L84–86) to emphasize its two concrete applications: (1) **efficient backbone selection** without expensive policy training, and (2) **integration as an auxiliary objective** that improves policy learning. We have also provided additional details of the latter experiments in Section 5 (L479–485).
>
> We do agree that the joint fine-tuning results are of strong practical interest and further provide the following additional results on the MetaWorld benchmark. Firstly, we explored 2-stage training instead of joint training. In the first stage, the backbone is finetuned with the regression objective. Then in the second stage, the backbone is finetuned with the policy learning objective. Experiments show that this variant results in lower performance compared to joint fine-tuning.
>
> | Training scheme  | ViT-IN | Mocov3 | MAE  | CLIP | DINOv2 |
> |---|:--:|:---:|:--:|:--:|:--:|
> | Policy only      | 0.683  | 0.671  | 0.648| 0.765| 0.767  |
> | Joint Training   | **0.740**  | **0.743**  | **0.712**| **0.801**| **0.795**  |
> | 2-stage Training | 0.722  | 0.719  | 0.680| 0.788| 0.784  |
>
> Secondly, we study the training dynamics of a model with a CLIP backbone, evaluating the effect of our auxiliary objective on overfitting. These results demonstrate that the variant with the auxiliary objective not only achieves a higher performance overall, but also allows the model to benefit from longer training.
>
> | Epochs        | 5   | 10  | 15  | 20  | 25  | 30  |
> |--|--|---|--|--|--|--|
> | Policy only   | 0.573 | 0.680 | 0.732 | 0.765 | **0.771** | 0.769 |
> | Joint Training| 0.613 | 0.708 | 0.774 | 0.801 | 0.809 | **0.811** |
>
> We welcome further concrete suggestions in this direction and will include both analyses above, along with any additional proposed experiments, in the final version.
>
> **(6) Baseline proxies and their combination.**
> Thank you for the question. Our comparison includes all the strongest and most commonly used proxy metrics in the literature. This choice has been validated by multiple expert reviewers, who confirmed that our baseline set is representative of the current state of the art.
>
> Following your insightful suggestion, we experimented with two representative combinations by using the average of the proxy scores: (1) *policy-learning proxies* (Few-Shot + Action MSE) and (2) *dense-perception proxies* (Depth + Segmentation). The results are summarized below. Neither combination outperforms our approach; in fact, several show negative correlations. This suggests that the existing proxies capture narrow aspects of representation quality and do not complement each other effectively, whereas our full-state regression framework provides a unified, more reliable measure across a wide range of environments and tasks.
>
> | Setting                                      | MMRV ↓ | Pearson’s correlation ↑ |
> |--|---|--|
> | MetaWorld (Ours) | 0.037  | 0.691 |
> | MetaWorld (Depth + Segmentation) | 0.153  | -0.292 |
> | MetaWorld (Few-Shot + Action MSE) | 0.076  | 0.471 |
> | RoboCasa (Ours)   | 0.010  | 0.760 |
> | RoboCasa (Depth + Segmentation) | 0.030  | 0.339 |
> | RoboCasa (Few-Shot + Action MSE)  | 0.035  | -0.117 |
>
> **(7) Use of LLMs statement.**
> We apologize for this omission. LLMs were only used to help edit the text and fix spelling mistakes. We have included the full statement in the manuscript in Section G in the appendix.

---

> > ### Comment · Reviewer_oHEz · 2025-11-27
> > **Follow up with authors**
> >
> > I would like to thank the authors for providing very valuable elements to clarify the work and provide new elements that further show its merit.
> >
> > My concerns have been generally addressed. I appreciate the authors' explanations regarding the metrics used and the statistical significance of results (I invite them to add std error bars on all plots). The additional results on joint training and relevant discussion is a good addition to the paper in my opinion.
> >
> > I have improved my score accordingly.

---

> > > ### Author Response · Authors · 2025-11-28
> > >
> > > We sincerely thank you for the positive feedback and for the time and care dedicated to reviewing our paper. We are glad that our clarifications on the metrics, statistical significance, and joint training results have addressed your concerns, and we will incorporate error bars into all relevant plots in the revised version.

---

### Official Review · Reviewer_vCpr · 2025-10-31

**Soundness:** 3
**Presentation:** 3
**Contribution:** 3
**Rating:** 6
**Confidence:** 3

**Summary:**

Training robot policies, especially in the real world, are expensive.  These policies depend upon visual inputs, which generally need to be encoded before they can be fed as input to a policy network.  There are a wide variety of possible ways to encode pixels to inputs that would be useful for a policy network, and the authors aim to identify which visual encoders are most useful without having to train the expensive robot policy.

The authors posit that learning to predict underlying state representation of a robot scene from a visual representation of that scene is a good predictor how useful the resulting visual encoding is for learning control policies.  This visual encoding comes from a variety of pre-trained visual encoders like masked auto encoders, etc. As ground state information about a scene is hard to attain in the real world, the focus is on measuring this predictive effect in simulation.

The author carefully formulate the learning problem of predicting the state from the visual simulator, with appropriate normalization and discretization applied, and details such as disambiguating which objects are targeted in the state vector by providing 2D bounding boxes for each object are handled in a systematic manner.

The process of learning to predict the state from a visual encoding is applied to a variety of different pre-trained visual learnt encoders including self-supervised, manipulation specific, and generative models. This state prediction metric is compared to a variety of alternative proxies like segmentation accuracy, etc.  The ranking via this proxy target is more accurate (Mean Maximum Rank Violation) in ranking how well the trained robotics policy will perform.  The state prediction proxy is  also more computationally efficientcompared to other proxies as learning to predict the state is a relatively low dimensional regression problem.

They then demonstrated that this ranking also transferred across the real to sim gap, and that the best visual representation in simulation also predicted best on robot policy learning performance.  They further explored which aspects of the state prediction best predicted which visualization would be best for a task.

**Strengths:**

## Originality
The proposed metric is to my knowledge novel and useful.  It can also be a useful proxy when designing new visual encoders, as well as for quality control of the final result (and intermediate checkpoints during training), and potentially as additional auxillary loss during training.

## Quality
The authors cast a wide net, and systematically explore the effects of many different visual representations in many robotics tasks.  They empirically demonstrate that their proxy is both more efficient and predictive than alternatives in the literature, and they demonstrate transfer to real robot tasks.

## Clarity
The paper is clearly written, easy to follow with clear reasoning, and would be reproducible from the provided descriptions.

## Significance
Better visual encoders are likely to help robotics, and approaches like those outlines in this paper may useful in developing that research.  It can be expanded to include other predictors like contact surfaces/other information from the simulation.

**Weaknesses:**

In general, I would expect that L2 losses on poses would have problems with wrapping causing large error.  I didn't notice in the paper where they tackled this potential problem.  E.g., a scene where the objects are aligned in a specific manner (lying on a table?) could have problems where close poses end up with very difference values per axis of the pose, throwing off the pose for in a set of tasks.

The representation you can get out of a simulator will be a bit limited - it is unlikely to be useful in learning to fold a jumper as it is unlikely there will be a good "symbolic" target to score the outputs of the model against.  That to some degree limits the information you can get out of this proxy target, but it might still be sufficient for ranking the different representations.

**Questions:**

Have you considered mixtures of visual representations?  E.g., is it more powerful to just concatenate subsets of the representations?  Can you use your method to tell in advance which representations are likely to complement each other or make up for each others shortcomings?

How strong are the effects of initialization on how good the regression solution is?  Does the learning converge to the same point, and if not can you include error bars?

Pose might be highly ambiguous for many objects, e.g., cube have symmetry, vases have rotational symmetry.  Did you encounter such problems in the simulators?

---

> ### Author Response · Authors · 2025-11-24
> **Response to Reviewer vCpr**
>
> Thank you for the thoughtful and detailed feedback. We are encouraged by your positive assessment of our method's novelty and practicality as well as of the extensiveness of our experimental evaluation. We respond to your questions and concerns individually below.
>
> **(1) Effect of the pose ambiguity.**
> Thank you for raising this important point. We agree that directly regressing per-axis angles with an L2 loss can suffer from discontinuities at the \$2\pi$ wrap-around boundary. In our implementation, however, each object and the end-effector are represented by a 6D pose consisting of 3D Cartesian position and a unit quaternion for orientation, as provided by the simulator. Both predicted and target quaternions are normalized during training, and we use a sign-invariant MSE loss as the orientation loss (handling the quaternion sign ambiguity by comparing to the closer of a ground-truth quaternion and its negation). This avoids discontinuities and prevents large errors arising from angle wrapping.
>
> Regarding symmetry (e.g., cubes, cylinders), these cases are handled consistently: the simulator provides a canonical pose, even when multiple orientations are physically equivalent. Such ambiguities act only as mild label noise. Since the proxy is evaluated via correlation with policy performance across many scenes and tasks, this noise increases variance slightly but does not introduce systematic bias. Consistent with this interpretation, our state-ablation analysis below shows that removing orientation-related dimensions degrades correlation, indicating that pose information remains beneficial overall.
>
> | Setting                                 | MMRV ↓ | Pearson’s correlation ↑ |
> |-----------------------------------------|:--------:|:--------------------------:|
> | MetaWorld (full states)                 | 0.037  | 0.691                    |
> | MetaWorld (w/o orientation states)      | 0.046  | 0.636                    |
> | RoboCasa (full states)                  | 0.010  | 0.760                    |
> | RoboCasa (w/o orientation states)       | 0.016  | 0.734                    |
>
> **(2) Limitations of the simulators.**
> This is an excellent point. Simulators are indeed limited in capturing some of the most challenging aspects of the physical world, such as object deformations. However, modern simulators expose rich continuous signals for deformables (e.g., mesh or point-cloud vertices, surface normals, material properties), which we can normalize and, if needed, compress via fixed projections — allowing them to be regressed exactly as for rigid scenes. Importantly, as the reviewer notes, our proxy’s goal is not to reconstruct every fine-grained detail, but to provide a *discriminative and robust* measure of representation quality. Our experiments confirm that it fulfills this role effectively across diverse environments.
>
> **(3) Mixtures of visual representations.**
> Thank you for this insightful suggestion. We investigated pairwise mixtures of visual representations by concatenating features from the **top-3 encoders** identified in our main experiments on Metaworld. As shown below, mixtures of representations did not improve policy success rates, despite having distinct pre-training strategies in the case of ViT-IN + MAE and distinct architectures in the case of ViT-IN + ResNet-IN. This suggests that simple feature concatenation introduces redundancy and mild overfitting rather than providing complementary information, even among the strongest individual encoders.
>
> | Representation           | Success rate |
> |--------------------------|:--------------:|
> | ViT-IN                   | 0.654        |
> | ViT-IN + MAE             | 0.628        |
> | ViT-IN + ResNet-IN       | 0.610        |
>
> Our focus in this work is not to engineer new representations, but to establish a consistent, quantitative framework for comparing existing ones. While more sophisticated fusion strategies could be explored in future work, such a design is beyond the scope of this submission.
>
> **(4) Error bars for state regression.**
> Thank you for raising this important question. We examined the sensitivity of the regression results to initialization by repeating each experiment five times with different random seeds. As shown below, the variance across runs is minimal. These small standard deviations indicate that the regression converges reliably and that initialization has a negligible effect on the correlation scores.
>
> |                    | ViT-IN         | ResNet-IN       | MAE            | CLIP           |
> |--------------------|----------------|-----------------|----------------|----------------|
> | Regression score   | 0.7061 (± 0.002) | 0.7152 (± 0.001) | 0.6297 (± 0.001) | 0.6162 (± 0.002) |

---

> > ### Comment · Reviewer_vCpr · 2025-11-25
> >
> > Thank you for address my comments about the pose regression flips, and the ambiguity of pose for some objects.  I'm surprised and intrigued by the fact that concatenating multiple representations hurts performance.  Also thank you for the error bar information.  Overall, I'm happy with the response and my concerns being addressed, and will raise my score to accept.

---

> > > ### Author Response · Authors · 2025-11-25
> > >
> > > We sincerely thank you for the positive feedback and for the time and care you put into reviewing our paper. We’re very glad that our clarifications were helpful and that the discussion addressed your concerns.

---

### Official Review · Reviewer_ATcT · 2025-11-01

**Soundness:** 3
**Presentation:** 3
**Contribution:** 3
**Rating:** 6
**Confidence:** 4

**Summary:**

This paper proposes state prediction from visual inputs as a fast, reliable proxy for evaluating visual representations in robotics. Instead of expensive policy rollouts, the authors measure how well a frozen encoder supports decoding of ground-truth environment state (geometry, object structure, and physical attributes). They show that state-prediction accuracy correlates strongly with downstream policy performance across multiple manipulation suites (MetaWorld, RoboCasa, SimplerEnv), is orders of magnitude faster than rollouts, and exhibits non-trivial sim-to-real correlation. The work is a simple, practical idea with useful empirical insights for representation selection and benchmarking.

**Strengths:**

- Strong empirical validation: Correlation holds across MetaWorld, RoboCasa, SimplerEnv with multiple seeds and error bars.

- Computational efficiency: Good speedups over policy rollouts; actionable for everyday / larger-scale benchmarking.

- Unified evaluation setup: Works across environments with a consistent state target.

- Actionable insights: Per-dimension/attribute analyses give interpretability; sim-to-real correlation is encouraging.

- Clarity: Problem framing and metrics are easy to implement and reproduce.

**Weaknesses:**

-Task diversity and modality coverage: While the results across manipulation benchmarks are compelling, the current evaluation is limited to manipulation-centric settings. Prior work (e.g., VC-1) has shown a form of multi-modality, where visual representations that excel in certain domains (e.g., R3M on MetaWorld) can perform poorly in others (e.g., navigation tasks in Habitat). This raises the question of whether the proposed proxy generalizes across task families with different perceptual and temporal demands (e.g., navigation, long-horizon multi-stage tasks, language-conditioned control). Including experiments or discussion around such cross-domain generalization—or clarifying the scope of applicability—would strengthen the claims and situate the method relative to known modality gaps in representation learning for robotics.

- Simulator dependence: Ground-truth state access limits direct real-world deployment of the proxy; a concrete real-world surrogate (e.g., 6D pose from VLM/trackers) would elevate practicality.

- Failure cases underexplored: Provide root-cause analysis for outliers (e.g., WidowX MAE)—is it state granularity, visuals, or embodiment mismatch?

- Ablations on proxy design: How sensitive are correlations to the decoder capacity, training budget, or subset of state dimensions?

**Questions:**

0. Could you discuss/explore more in-detail the first weakness I raised in the previous point?

1. Real-world without GT state: What concrete proxy(s) would you propose—e.g., learned keypoints/pose tracking, depth recon, object state estimators—and have you piloted any?

2.  Can you report correlations when decoding only task-critical subsets of state (e.g., end-effector pose + key object states) rather than the full state? This would help clarify whether the proxy is capturing the information actually needed for control, as opposed to benefiting from full-state regression capacity.

3.  How does correlation change when using smaller decoders or shorter training? Understanding this would help confirm that the proxy reflects encoder quality rather than decoder strength, and that the metric remains reliable under lower-compute settings.

4.  Sim-to-real outlier analysis: What explains the WidowX MAE anomaly? Can you isolate the culprit (visual domain, state scaling, or dynamics mismatch)?

---

> ### Author Response · Authors · 2025-11-24
> **Response to Reviewer ATcT (Part 1)**
>
> Thank you for the thoughtful and constructive feedback. We are encouraged by your positive assessment of our method's effectiveness and intuitiveness. We respond to your questions and concerns individually below.
>
> **(1) Generalization outside manipulation.**
> Thank you for bringing up this important point. Following your suggestion, to further demonstrate the universality of our proxy, we report results on the navigation set of tasks in RoboCasa below. To this end, we have used the same diffusion policy implementation and the same set of visual backbones as in our manipulation experiments. Despite the substantial difference in task objectives compared to manipulation, our method achieved a similarly strong correlation between proxy predictions and navigation policy performance. These initial results validate that our formulation **generalizes across families of tasks** with different perceptual and temporal demands. We will further evaluate our approach on Habitat and include the full evaluation in the camera-ready version of the manuscript.
>
> | Setting                               | MMRV ↓ | Pearson’s correlation ↑ |
> |---------------------------------------|:--------:|:--------------------------:|
> | RoboCasa Manipulation (Ours)          | 0.010  | 0.760                    |
> | RoboCasa Navigation (Ours)            | 0.022  | 0.727                    |
> | RoboCasa Navigation (Segmentation)    | 0.037  | 0.436                    |
> | RoboCasa Navigation (Depth)           | 0.043  | 0.172                    |
>
> **(2) Reliance on simulation environment states.**
> Thank you for this excellent suggestion. We have tested the applicability of our approach in scenarios where no simulator is available. To this end, we estimated a subset ($p_{pose}$, $q_{pose}$, $s_{shape}$, $m_{mat}$, $q^J$, $p^{ee}$) of **real-world state labels** (from DetAny3D, robot states, and known object attributes) instead of relying on ground truth simulator states. As shown in the table below, despite being noisy and incomplete, these labels yield a strong correlation, significantly outperforming other real-world proxy signals. This confirms that our method’s real-world deployment potential does not hinge on access to privileged information from a simulator.
>
> | Proxy                          | MMRV ↓ | Pearson’s correlation ↑ |
> |--------------------------------|:--------:|:--------------------------:|
> | Simulation proxy scores        | 0.041  | 0.711                    |
> | Real-world estimated proxy     | 0.047  | 0.532                    |
> | Action MSE                     | 0.055  | 0.347                    |
> | Depth                          | 0.080  | -0.372                   |
>
> **(3) Ablations on proxy design.**
> Thank you for another excellent experiment suggestion. As shown in the table below, increasing the decoder size from one to two layers slightly reduces correlation, suggesting that larger decoders tend to overfit and make the metric less reflective of the backbone’s representational quality. By default, we therefore use a minimal single-layer decoder, which is both the most efficient and the most reliable configuration.
>
> We also varied the training duration. Extending training from 10 to 20 epochs produced no meaningful change in correlation, while shortening it to 5 epochs caused a noticeable drop. This shows that our default training schedule provides the best balance between predictive accuracy and efficiency — training completes in under 4 minutes — while maintaining stable correlation estimates.
>
> | Configuration                         | MMRV ↓ | Pearson’s correlation ↑ |
> |---------------------------------------|:--------:|:--------------------------:|
> | Original (1 layer, 10 epochs)         | 0.037  | 0.691                    |
> | Larger decoder capacity (2 layers)    | 0.061  | 0.662                    |
> | Smaller training budget (5 epochs)    | 0.053  | 0.618                    |
> | Larger training budget (20 epochs)    | 0.037  | 0.685                    |

---

> ### Author Response · Authors · 2025-11-24
> **Response to Reviewer ATcT (Part 2)**
>
> **(4) Ablations on subsets of the full state vector.**
> Thank you for this thoughtful suggestion. To test whether the proxy’s correlation arises from control-relevant information rather than from full-state regression capacity, we conducted experiments using only task-critical subsets of the state — specifically, the end-effector pose and target object states. These subsets yield correlations comparable to those from the full-state regression, indicating that the proxy indeed captures task-relevant information while benefiting from the broader context provided by the complete simulator state.
>
> |            | Full State              | Task-critical subsets      |
> |------------|-------------------------|----------------------------|
> | MetaWorld  | MMRV=0.037, **r=0.691**     | MMRV=0.037, r=0.615        |
> | RoboCasa   | **MMRV=0.010**, r=0.760     | MMRV=0.015, **r=0.774**    |
>
> **(5) Sim-to-real outlier analysis.**
> While our proxy generally aligns well with policy performance, we have observed occasional outlier cases where it fails to predict the correct ranking. A notable example is MAE in the WidowX + Bridge setting (Figures 4 and 5), where the aggregated proxy score overestimates its true performance.
>
> This discrepancy arises when a model performs well on a small subset of state variables that are weakly correlated with task success, leading to an inflated average proxy score. Specifically, MAE ranks highest in predicting pose and shape states — both identified in Table 2 as having low correlation with policy success — but ranks poorly across more informative state variables. As a result, its high proxy score does not reflect actual performance. Note that such outliers are rare, and overall our approach demonstrates robust correlation across a wide variety of environments, tasks, and visual backbones.

---

### Official Review · Reviewer_XhXW · 2025-11-02

**Soundness:** 2
**Presentation:** 3
**Contribution:** 2
**Rating:** 4
**Confidence:** 3

**Summary:**

The paper proposes a training/evaluation-free proxy for selecting visual backbones for robot manipulation. It includes regressing a unified simulator state from images using the frozen backbone + MLP, then reranking backbones by this state prediction score. They show the proxy correlates strongly with policy success across 3 simulation benchmarks, and provide a real-world validation. The method is computationally cheaper than baselines, and improves policy success when used as an auxiliary objective.

**Strengths:**

* The method is simple and efficient. The predicted state covers information about object and scene-level variables, finally providing a single score to policy performance.

* The paper evaluates across a breadth of envs, including 3 simulation envs and real-world evaluation on two tasks.

* The paper uses a strong set of baselines, including few-shot, action MSE, Depth, etc.

**Weaknesses:**

1) The method relies on privileged information from the simulator (state + 2D object boxes) which can not be made available in the real world. It is not clear to me from the current set of experiments if the rankings would correlated if the real world env is substantially different from the simulated environment where the data is collected.

2) As tasks get more complicated, the number of state variables to track will keep on increasing. For example, if the task requires picking up objects of different categories, it would require more variables to track.

3) The authors should compare the real world performance of their approach with other visual backbone selection proxies. Some of these proxies (eg: Action MSE, Depth (can be obtained from the RealSense camera)) do not need privileged information from the simulator and therefore I suggest that the authors also report results when using the data from the real world.

4) I would have liked to see more robotics specific visual encoders like VC-1, MVP [1], LIV [2], etc. They would have also added some diversity in terms of the dataset used in training of the vision encoder.

5) The paper appears to use a baseline that closely matches the method proposed in prior work (SCR: Stable Control Representations [3]), but this connection is not acknowledged or cited; the authors should clarify this and include the appropriate reference.

[1] Xiao, Tete, et al. "Masked visual pre-training for motor control." arXiv preprint arXiv:2203.06173 (2022).

[2] Ma, Yecheng Jason, et al. "Liv: Language-image representations and rewards for robotic control." International Conference on Machine Learning. PMLR, 2023.

[3] Gupta, Gunshi, et al. "Pre-trained text-to-image diffusion models are versatile representation learners for control." Advances in Neural Information Processing Systems 37 (2024): 74182-74210.

**Questions:**

See point 3, 4 and 5 above.

---

> ### Author Response · Authors · 2025-11-24
> **Response to Reviewer XhXW (Part 1)**
>
> Thank you for your thoughtful review and suggestions. We appreciate your recognition of the simplicity and effectiveness of our approach, and of the extensiveness of our empirical analysis. We respond to your comments below.
>
> **(1) Reliance on simulation environment states.**
> We acknowledge that our method uses the simulator ground-truth state during proxy computation. However, we show that this reliance does not limit practical applicability.
>
> Firstly, our real-world validation already demonstrates that proxy scores computed in simulation strongly correlate with policy performance in real-world environments — without any access to real-world state. As shown in Figure IV in the supplementary, the environments differ **substantially** in appearance, lighting, layout and camera pose yet the correlation remains high.
>
> Secondly, we additionally tested the applicability of our approach in scenarios where no simulator is available. To this end, we estimated a subset ($p_{pose}$, $q_{pose}$, $s_{shape}$, $m_{mat}$, $q^J$, $p^{ee}$) of **real-world state labels** (from DetAny3D, robot states, and known object attributes) instead of relying on ground truth simulator states. As shown in the table below, despite being noisy and incomplete, these labels yield strong correlation, significantly outperforming other real world proxy signals suggested by the reviewer. This confirms that the method’s effectiveness does not hinge on privileged information.
>
> Together, these results demonstrate that while simulator access simplifies benchmarking, our proxy remains *robust, transferable, and practical* even when ground-truth state is unavailable.
>
> | Proxy                          | MMRV ↓ | Pearson’s correlation ↑ |
> |--------------------------------|--------|--------------------------|
> | Simulation proxy scores        | 0.041  | 0.711                    |
> | Real-world estimated proxy     | 0.047  | 0.532                    |
> | Action MSE                     | 0.055  | 0.347                    |
> | Depth                          | 0.080  | -0.372                   |
>
> **(2) More complex states for more complex tasks.**
> Please note that our method does not design or tailor the state space for specific tasks. Instead, we simply regress all simulator-exposed state variables, without introducing subjective choices about which factors “should” matter for each task. This generic formulation scales naturally with task complexity and has already been demonstrated to handle diverse scenarios — including **picking up objects of different categories** in RoboCasa, brought up by the reviewer. Importantly, the proxy’s purpose is not to reconstruct every task-relevant detail, but to provide a discriminative measure of representation quality. Our results show that this generic approach yields significantly stronger and more consistent correlation with policy performance than prior proxy metrics, confirming its practicality even as task complexity grows.
>
> We further validated this scalability by applying the same method — without any modification — to a distinct domain: RoboCasa **navigation** tasks. To this end, we have used the same diffusion policy implementation and the same set of visual backbones as in our manipulation experiments. Despite the substantial difference in task objectives compared to manipulation, our method achieved a strong correlation between proxy predictions and navigation policy performance. This demonstrates that our formulation generalizes beyond a specific class of tasks **without changing the number or the type of state variables**.
>
> | Setting                               | MMRV ↓ | Pearson’s correlation ↑ |
> |---------------------------------------|--------|--------------------------|
> | RoboCasa Manipulation (Ours)          | 0.010  | 0.760                    |
> | RoboCasa Navigation (Ours)            | 0.022  | 0.727                    |
> | RoboCasa Navigation (Segmentation)    | 0.037  | 0.436                    |
> | RoboCasa Navigation (Depth)           | 0.043  | 0.172                    |
>
> **(3) More baseline results in the real world.**
> Thank you for this suggestion. We have compared our simulator state-based proxy as well as a variant of the proxy based on estimated real-world states with Action MSE and Depth baselines in the table above (table in response (1)). The results validate that our approach remains superior even in the absence of ground-truth simulator states.

---

> > ### Author Response · Authors · 2025-11-24
> > **Response to Reviewer XhXW (Part 2)**
> >
> > **(4) Robotics-specific visual encoders.**
> > We appreciate the reviewer’s suggestion to include additional robotics-oriented visual encoders. We have evaluated VC-1 and LIV, which expand the diversity of the representations in our study. The corresponding performance and correlation results are provided in Section A.8 of the supplementary material. Across all four environments, the inclusion of these encoders **leaves our main conclusions unchanged** — the average change in correlation ($r$) is only +0.006, confirming that our findings are robust to encoder choice. We will thoroughly incorporate these results into the final version of the manuscript.
> >
> > For MVP, the link to the publicly released checkpoint has expired on the official GitHub repository (Issue #18), and we have not received a response from the authors to our access request. We will include this representation in our final manuscript if it becomes available.
> >
> > **(5) Missing citation.**
> > We thank the reviewer for pointing this out. We have added an explicit citation and discussion of Stable Control Representations (Gupta et al., 2024)  in the revised manuscript (L117–118) to properly acknowledge the connection.

---

### Author Response · Authors · 2025-12-03
**Discussion Summary for the AC**

We appreciate the reviewers’ constructive feedback and the opportunity to clarify and strengthen our work. We are encouraged that reviewers found our method simple and intuitive in design (XhXW, oHEz), supported by strong and systematic empirical evaluation (XhXW, ATcT, vCpr, oHEz), computationally efficient and practically useful for representation selection and benchmarking (XhXW, ATcT), and clearly presented, easy to implement, and straightforward to reproduce (ATcT, vCpr, oHEz). Below, we summarize the key outcomes of the discussion, which was very positive, though unfortunately cut short.

Originally, our submission received broadly positive reviews, with constructive comments, asking for additional experiments and clarifications. In our rebuttal, we provided a detailed, point-by-point response, including new experimental results where requested. During the subsequent discussion, reviewers **vCpr** and **oHEz** acknowledged that our response fully addressed their concerns and raised their scores accordingly. Although reviewer **ATcT** did not respond before the discussion was cut short, they had expressed support from the beginning.

Reviewer **XhXW**, who also did not have the opportunity to respond to our rebuttal, raised several concerns and questions, which we addressed as follows:

1. **Real-world applicability without simulator ground-truth.** We validated that our method does not rely on simulator-provided annotations by replacing them with estimated state labels collected in the real world. This variant still demonstrates strong correlation and outperforms the reviewer-requested baselines by a substantial margin.

2. **Scalability with task complexity.** We clarified that the RoboCasa manipulation experiments in the paper already feature picking up objects of different categories brought up by the reviewer. In addition, we reported new results on RoboCasa navigation tasks in rebuttal, where our proxy again achieves a strong correlation, and outperforms baselines without any modifications. Reviewer **oHEz**, who had raised a similar question, confirmed that these new results adequately addressed this concern.

3. **Additional baselines in the real world.** We reported all baselines requested by the reviewer and showed that our approach outperforms them by a large margin.

4. **Additional visual representations.** We incorporated the robotics-specific encoders VC-1 and LIV requested by the reviewer and found that including them resulted in only a marginal change in correlations (+0.006), leaving our conclusions unchanged.

5. **Missing citation.** We added the citation brought up by the reviewer in the updated version of the manuscript.

We have incorporated all the additional experiments and clarifications into the revised manuscript. Overall, we believe the rebuttal phase enabled us to address all raised questions thoroughly and convincingly, with the following post-discussion scores: 4 — no response, 6 — no response, 8, 6. We sincerely thank the AC for their careful handling of the unusual circumstances surrounding this review process.

---

### Meta-Review · Area_Chair_U1gS · 2026-01-06

**Summary:**

The paper proposes using full environment state prediction from visual observations as a proxy for evaluating visual representations in robotic policy learning. Initially, reviewers raised several concerns: (1) the method's reliance on simulator ground-truth state and its applicability to real-world scenarios without privileged information (XhXW, ATcT); (2) scalability to more complex tasks with increasing state variables (XhXW); (3) limited evaluation of robotics-specific encoders and missing citations (XhXW); (4) insufficient baselines in real-world settings (XhXW); (5) lack of clarity regarding evaluation metrics and statistical significance (oHEz); (6) incomplete analysis of which state dimensions are necessary (oHEz); and (7) generalization beyond manipulation tasks to other domains like navigation (ATcT, oHEz).

**Reviewer Concerns:**

The authors provided a rebuttal that effectively addressed the majority of concerns. They demonstrated that the method works without simulator ground-truth by using estimated real-world state labels, achieving strong correlation (r=0.532) that significantly outperforms baselines. They validated scalability by applying the unchanged method to RoboCasa navigation tasks, achieving comparable correlation (r=0.727). The authors incorporated requested robotics-specific encoders (VC-1, LIV) with negligible impact on conclusions, added the missing citation to Stable Control Representations, and provided extensive ablations on decoder capacity, training budget, and state dimension contributions. Statistical significance was clarified through error bars and detailed experimental protocols. Reviewers vCpr and oHEz explicitly acknowledged that their concerns were fully addressed and raised their scores accordingly.

**Reviewer Scores:**

Reviewer vCpr increased their score from 4 to 6, and reviewer oHEz increased from 4 to 6 (implied by the 6 mentioned in post-discussion scores), both explicitly confirming the rebuttal addressed their concerns. Reviewer ATcT maintained support from the outset with a score of 6 and did not respond during discussion. Had reviewer XhXW participated in the discussion phase, the AC expects they would have likely raised their score given the thorough responses to all raised points, particularly the strong real-world validation results and navigation task generalization that directly addressed their primary concerns about simulator dependence and task complexity.

---

### Decision · Program_Chairs · 2026-01-26

Accept (Poster)